# Cooperative base pair melting by helicase and polymerase positioned one nucleotide from each other

**Divya Nandakumar, Manjula Pandey, Smita S Patel***

Department of Biochemistry and Molecular Biology, Rutgers-Robert Wood Johnson Medical School, Piscataway, United States

**Abstract** Leading strand DNA synthesis requires functional coupling between replicative helicase and DNA polymerase (DNAP) enzymes, but the structural and mechanistic basis of coupling is poorly understood. This study defines the precise positions of T7 helicase and T7 DNAP at the replication fork junction with single-base resolution to create a structural model that explains the mutual stimulation of activities. Our 2-aminopurine studies show that helicase and polymerase both participate in DNA melting, but each enzyme melts the junction base pair partially. When combined, the junction base pair is melted cooperatively provided the helicase is located one nucleotide ahead of the primer-end. The synergistic shift in equilibrium of junction base pair melting by combined enzymes explains the cooperativity, wherein helicase stimulates the polymerase by promoting dNTP binding (decreasing dNTP $K_m$), polymerase stimulates the helicase by increasing the unwinding rate-constant ($k_{cat}$), consequently the combined enzymes unwind DNA with kinetic parameters resembling enzymes translocating on single-stranded DNA.

## Introduction

Replicative helicases and DNA polymerases (DNAPs) are not efficient at unwinding the duplex DNA when they are working independently. The unwinding rates are slower than their translocation rates on single-stranded (ss) DNA and slower than the rates of DNA replication (*Kim et al., 1996*; *Delagoutte and von Hippel, 2001*; *Galletto et al., 2004*; *Stano et al., 2005*; *Lionnet et al., 2007*; *Donmez and Patel, 2008*). Moreover, the unwinding rates of isolated helicases decrease steeply with increasing GC percentage in the duplex DNA, therefore, assisting forces that destabilize the junction base pairs stimulate the helicase (*Galletto et al., 2004*; *Johnson et al., 2007*; *Lionnet et al., 2007*; *Donmez and Patel, 2008*). However, in the presence of an actively synthesizing replicative DNAP, the unwinding rates of the helicase become fast and GC independent (*Kim et al., 1996*; *Delagoutte and von Hippel, 2001*; *Stano et al., 2005*; *Manosas et al., 2012b*; *Pandey and Patel, 2014*). Similarly, replicative DNAPs on their own have limited strand displacement synthesis activity, restricted to 4–6 base pairs in T7 DNAP (*Stano et al., 2005*; *Yuan and McHenry, 2009*; *Pandey and Patel, 2014*). The isolated DNAPs often stall and move backward to excise the nascent DNA using their proofreading exonuclease activity when faced with downstream duplex DNA (*Manosas et al., 2012a*). The presence of helicase or single-stranded DNA-binding protein (SSB) inhibits processive excision of the nascent DNA and allows DNAP to catalyze fast and processive strand displacement synthesis (*Manosas et al., 2012b*; *Pandey and Patel, 2014*). Originally identified in prokaryotic systems (T7, T4 bacteriophage, *Escherichia coli*), this functional coupling between the helicase and DNAP is found also in eukaryotic replication systems (*Kang et al., 2012*).

Several models have attempted to explain the functional coupling between replicative helicases and DNAPs. The underlying theme of the helicase only unwinding models is that the helicase unwinds

**\*For correspondence:** patelss@rutgers.edu

**Competing interests:** The authors declare that no competing interests exist.

**eLife digest** DNA replication is the process whereby a molecule of DNA is copied to form two identical molecules. First, an enzyme called a DNA helicase separates the two strands of the DNA double helix. This forms a structure called a replication fork that has two exposed single strands. Other enzymes called DNA polymerases then use each strand as a template to build a new matching DNA strand.

DNA polymerases build the new DNA strands by joining together smaller molecules called nucleotides. One of the new DNA strands—called the 'leading strand'—is built continuously, while the other—the 'lagging strand'—is made as a series of short fragments that are later joined together. Building the leading strand requires the helicase and DNA polymerase to work closely together. However, it was not clear how these two enzymes coordinate their activity.

Now, Nandakumar et al. have studied the helicase and DNA polymerase from a virus that infects bacteria and have pinpointed the exact positions of the enzymes at a replication fork. The experiments revealed that both the polymerase and helicase contribute to the separating of the DNA strands, and that this process is most efficient when the helicase is only a single nucleotide ahead of the polymerase.

Further experiments showed that the helicase stimulates the polymerase by helping it to bind to nucleotides, and that the polymerase stimulates the helicase by helping it to separate the DNA strands at a faster rate. The next challenge is to investigate the molecular setup that allows the helicase and polymerase to increase each other's activities.

the duplex DNA creating ssDNA template for the DNAP and the DNAP traps the displaced ssDNA through DNA synthesis (*Delagoutte and von Hippel, 2001*; *Stano et al., 2005*). Recent studies of bacteriophage T7 and T4 DNAPs suggest alternative models indicating that DNAP aids the helicase by destabilizing the first few base pairs of the double-stranded (ds) DNA (*Manosas et al., 2012a*, *2012b*). Exonuclease mapping showed that T7 DNAP is located with the T7 helicase in close proximity to the fork junction and in a position to influence the junction base pairs (*Pandey and Patel, 2014*). To understand functional coupling between helicase and DNAP, we need to understand the basic mechanism of DNA unwinding by each enzyme. Although the mechanism of replicative DNAPs is well characterized on ssDNA template (*Patel et al., 1991*; *Doublie et al., 1998*; *Delagoutte, 2012*), there is little known about the mechanism of DNA unwinding-synthesis on duplex DNA template. Similarly, there are no structures of replisomes with the exception of a small angle X-Ray scattering structure of the T7 helicase-T7 DNAP bound to ssDNA and primer template DNA, respectively (*Kulczyk et al., 2012*). However, the low-resolution structure in the absence of a replication fork DNA does not provide the location of the DNAP and helicase at the replication fork junction to understand which enzyme is involved in melting the base pair at the fork junction.

In addition to structural questions such as the positioning of the helicase and DNAP at the replication fork, many aspects of the mechanism of functional coupling remain unexplored. For example, T7 and *E. coli* DNAPs are capable of strand displacement synthesis in the presence of SSB with rates comparable to their replication rates (*Yuan and McHenry, 2009*; *Pandey and Patel, 2014*). Hence, the specific role of the helicase in stimulating the synthesis activity of DNAP remains unclear. As combined enzymes, T7 helicase and T7 DNAP exhibit highly coordinated catalysis, whereby helicase hydrolyzes one dNTP for every dNMP incorporated by the DNAP (*Pandey and Patel, 2014*). This implies that the two enzymes coordinate their steps of nucleotide binding (2′-deoxythymidine 5′-triphosphate (dTTP) binding to T7 helicase and dNTP binding to T7 DNAP) and catalysis (dTTP hydrolysis and dNMP incorporation), but there is no model that explains how these steps are coupled between the two enzymes during active leading strand synthesis.

The studies in this paper use a combination of 2-aminopurine (2-AP) fluorescence and transient-state kinetics to investigate the unwinding mechanisms of T7 DNAP and T7 helicase as isolated enzymes and as combined enzymes. The kinetics indicates that DNAP and helicase use different mechanisms to unwind DNA, but the mechanisms when coupled generate an efficient replisome. In the replisome, T7 DNAP stimulates the helicase by increasing the unwinding $k_{cat}$, and T7 helicase stimulates the DNAP by decreasing the dNTPs $K_m$. The 2-AP studies probe DNA melting with single

base pair resolution and show that the isolated enzymes are not as efficient at melting the fork junction as compared to the combined enzymes. However, T7 DNAP with its ability to melt two base pairs ahead of the primer-end positions T7 helicase two nucleotides ahead with efficient and synergistic melting of the junction base pair. Overall, these studies provide both kinetic and structural basis to understand how helicase and polymerase mutually stimulate each other's activities during leading strand synthesis.

## Results

We have used 40 bp preformed replication fork substrates to measure the base pair unwinding rates using stopped-flow fluorescence based real-time assay. The replication fork substrate contains a $5'$-dT$_{35}$ tail that mimics the lagging strand, and DNA primer annealed to the $3'$-tail that mimics the leading strand (*Figure 1A*). We prepared a set of replication forks (*Supplementary file 1*-Table 1) where the sequence of the 40-base pair duplex was engineered to contain 20–65% GC content to study the effect of increasing resistance to movement on DNA-unwinding rates.

### DNA unwinding by T7 DNAP with *E. coli* SSB is rate-limited by base pair separation

To measure the DNA-unwinding kinetics in real time, we labeled the $3'$-end of the lagging strand with fluorescein and the $5'$-end of the leading strand with a black hole quencher (BHQ1) (*Figure 1A*). The fluorescence intensity of fluorescein-labeled lagging strand is quenched by BHQ1 when the DNA is duplexed, but when the lagging strand is unwound by T7 DNAP + *E. coli* SSB the fluorescence intensity increases (*Figure 1B*). We find that T7 DNAP on its own does not unwind the 40 bp replication fork at fast rates, which is consistent with our previously reported gel-based studies showing T7 DNAP stalling after 4–6 base pairs synthesis (*Stano et al., 2005*; *Pandey and Patel, 2014*) and this can be overcome by adding SSB (*Pandey and Patel, 2014*). Indeed, T7 DNAP fully unwinds the fork DNA with *E. coli* SSB in the reactions. Note that we can replace *E. coli* SSB with T7 gp2.5 (*Figure 1—figure supplement 1A*) (*Myers and Romano, 1988*; *Nakai and Richardson, 1988*; *Pandey and Patel, 2014*). However, we have used *E. coli* SSB because of its higher affinity for ssDNA (0.1–10 nM) (*Molineux et al., 1975*), requiring lower amounts of SSB for the same degree of stimulation as with T7 gp2.5 (200 nM SSB vs 3 µM T7 gp2.5) (*Figure 1—figure supplement 1B*). The *E. coli* SSB does not unwind DNA on its own (*Figure 1—figure supplement 1C*).

In these experiments, T7 DNAP and *E. coli* SSB were preassembled on the fork DNA, and DNA unwinding was initiated with dNTPs and Mg(II) in a stopped-flow instrument at 18°C. By preassembling the DNAP-SSB-DNA complex, we bypass the slow enzyme binding steps and synchronize the unwinding reactions. Therefore, the unwinding kinetics shows a presteady state time lag prior to fluorescence increase (*Figure 1B*). The lag time represents the time of unwinding, because it gets shorter when the duplex DNA to be unwound is 25 bp rather than 40 bp (*Figure 1—figure supplement 1D*). Therefore, the kinetics were fit to the *n*-step model (*Levin et al., 2009*) to extract the base pair unwinding rates (Appendix—Section 1 and *Figure 1—figure supplement 2A–C*). These average unwinding rates include time spent in unwinding the 40 bp fork DNA and time spent in any paused states.

The unwinding rates of T7 DNAP (with SSB) increase in a hyperbolic manner with increasing dNTPs concentrations with each of the GC forks (*Figure 1C*). At low-dNTPs concentration, the unwinding rates are GC-sensitive, but the rates reach a similarly high value at saturating dNTPs concentrations. Thus, the unwinding $k_{cat}$ remains nearly constant and ranges from 190 to 140 bp/s as GC percentage increases (*Figure 1D*). However, the dNTPs $K_m$ increases steeply from 40 µM to 270 µM as the GC percentage increases (*Figure 1E*). This indicates that DNA unwinding-synthesis by T7 DNAP (with SSB) is rate-limited by base pair separation at low-dNTPs concentrations, but not at high dNTPs. For comparison, when the downstream DNA is single-stranded and does not need to be melted, the dNTPs $K_m$ is 10–20 µM and the rate of synthesis is ~200 nt/s (*Patel et al., 1991*; *Stano et al., 2005*). Thus, the dNTPs $K_m$ on duplex DNA template is 2–10 times higher than on ssDNA template.

The minimal mechanism of DNA synthesis contains three steps (*Figure 1F*). The first step is capture of the templating base (base-capture) in the insertion site of the DNAP through translocation (Pol$_n$ ⇔ Pol$_n$.base), the second step is dNTP binding and base pairing with the templating-base (Pol$_n$.base ⇔ Pol$_n$.base.dNTP), and the third step is chemistry where dNMP is added, PPi is released, and the primer-end is elongated by one nucleotide (Pol$_n$.base.dNTP ⇔ Pol$_{n+1}$). The rate vs dNTPs

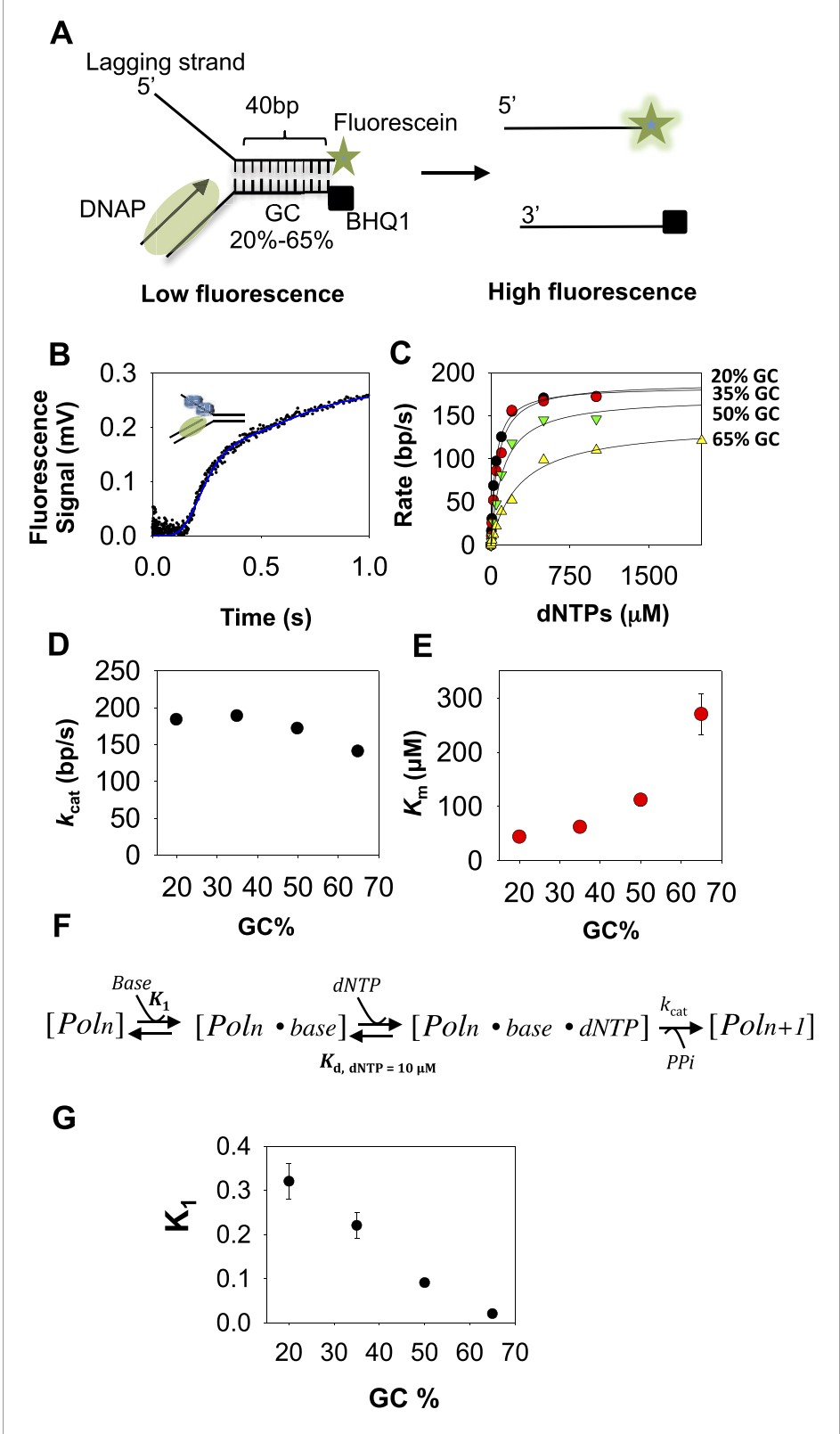

**Figure 1**. Kinetic mechanism of DNA unwinding by T7 DNAP. (**A**) Preformed replication fork DNA substrate for the measurement of the unwinding kinetics of T7 DNA polymerase (DNAP). (**B**) Representative kinetics of DNA unwinding by T7 DNAP in the presence of *E. coli* single-stranded DNA-binding protein (SSB) (dots) fit to the *n*-step model (solid line). (**C**) The base pair unwinding rates are plotted against dNTPs concentration and fit to *Equation 4*
*Figure 1. continued on next page*

*Figure 1. Continued*

(solid line) to obtain the unwinding $k_{cat}$ and dNTPs $K_m$ values. (**D**, **E**) The unwinding $k_{cat}$ and $K_m$ of T7 DNAP plotted against GC percentages. (**F**) The three-step ordered model of base-capture, dNTP binding, and nucleotide incorporation that describes the unwinding kinetics of T7 DNAP. (**G**) Equilibrium constants of the base-capture step obtained from fitting the kinetics data to the three-step model in **F** are plotted against increasing GC percentages. Details of the kinetic fittings are provided in Appendix—Section 2.

The following figure supplements are available for figure 1:

**Figure supplement 1**. Unwinding by T7 DNAP with gp2.5 and SSB.

**Figure supplement 2**. Unwinding trace of T7 DNAP with SSB.

**Figure supplement 3**. Fitting kinetics of unwinding by T7 DNAP with SSB.

dependencies of all GC forks fit well to this ordered three-step mechanism (Appendix—Section 2 and *Figure 1—figure supplement 3*) with variable equilibrium constant for the base-capture step ($K_1$), which we find decreases steadily with increasing GC percentage (*Figure 1G*). This means that the downstream DNA stability mainly affects the base-capture step, and DNAP is less efficient at capturing the templating-base when the stability of the duplex DNA is higher. However, the nearly GC-independent DNA unwinding $k_{cat}$ indicates that dNTP binding stabilizes the base-captured state and drives DNA unwinding, but higher concentrations of dNTPs are required with higher GC percentage in the downstream DNA. This mechanism of DNAP resembles the mechanism proposed for DNA-dependent RNA polymerases (*Thomen et al., 2005*).

## DNA unwinding by T7 helicase alone is rate-limited by base pair separation

The unwinding rates of T7 helicase were measured using the same replication fork substrates, except fluorescein was moved to the end of the leading strand (to avoid fluorescence changes from helicase binding to the end of the lagging strand) and a GGG quencher was introduced opposite to the fluorescein in the lagging strand to ensure that the signal is quenched when the DNA is duplexed (*Figure 2A*). The fork DNA was incubated with T7 helicase in the presence of dTTP without Mg(II) to avoid assembly lags, and the unwinding reactions were initiated with Mg(II) and $dT_{90}$ trap DNA. The unwinding kinetics show a presteady state time lag followed by fluorescence increase due to strand separation (*Figure 2B*). The kinetics were fit to the *n*-step model, and the average base pair unwinding rates were determined at various dTTP concentrations with all the GC forks (*Figure 2C*).

Unlike T7 DNAP, the unwinding $k_{cat}$ of the helicase decreases from 65 bp/s to 15 bp/s with increasing GC percentage (*Figure 2D*), but the dTTP $K_m$ remains relatively unchanged and decreases only two-fold from 300 µM to 160 µM (*Figure 2E*). Thus, T7 helicase and T7 DNAP respond differently to increasing GC content, which indicates that their DNA-unwinding mechanisms are fundamentally different.

T7 helicase moves on DNA through sequential nucleotide hydrolysis and translocation mechanism, where each subunit of the ring takes turn in binding the incoming nucleotide (*Liao et al., 2005*; *Crampton et al., 2006*; *Enemark and Joshua-Tor, 2006*; *Thomsen and Berger, 2009*; *Patel et al., 2011*). Therefore at any given time, only the leading subunit of T7 helicase binds an incoming dTTP and reels in the nucleotide base from the fork junction (*Sun et al., 2011*). There are three minimal steps during each base pair unwinding event. One, the leading helicase subunit binds to the nucleotide base from the fork junction (base-capture); two, the leading subunit binds to a molecule of dTTP; and three, dTTP hydrolysis and product release (2′-deoxythymidine 5′-diphosphate (dTDP) and Pi) occur at distinct subunits around the hexameric ring (*Figure 2F*). The order of events at the leading subunit is not known. In other words, it is not known whether base-capture by the leading subunit occurs in the dTTP-bound or dTTP-free state. Therefore, we considered all three models and our kinetic modeling shows that the rate vs dTTP data do not fit to models where dTTP binding and base-capture steps occur in a particular order (Appendix—Section 3 and *Figure 2—figure supplement 1A–C*). Instead, the data fit well to the random order mechanism (*Figure 2F* and *Figure 2—figure supplement 1D*), wherein base-capture can

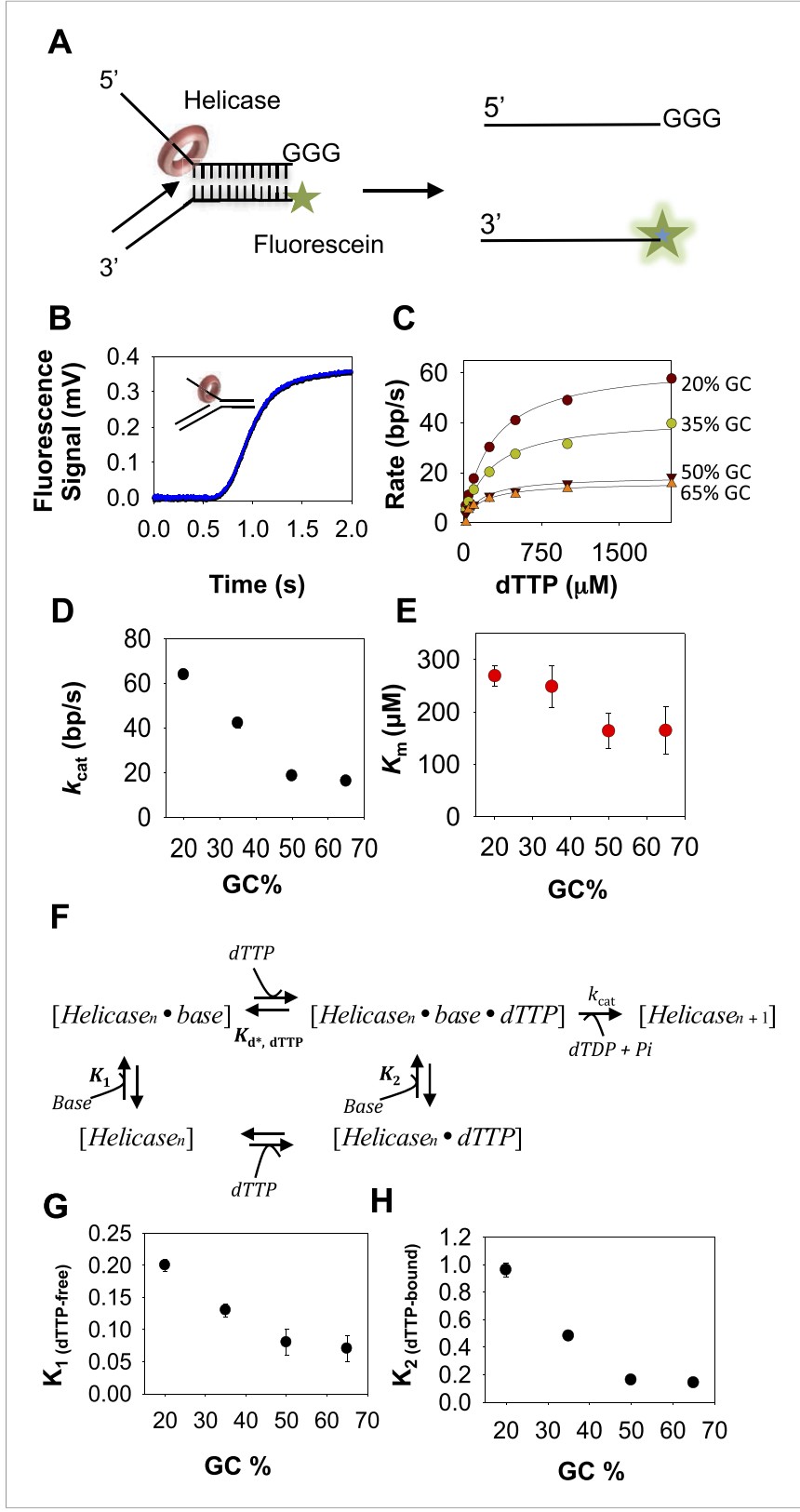

**Figure 2**. Kinetic mechanism of DNA unwinding by T7 helicase. (**A**) Replication fork DNA for the measurement of the unwinding kinetics of T7 helicase. (**B**) Representative kinetic trace of DNA unwinding by T7 helicase (dots) fit to the *n*-step model (solid line). (**C**) The unwinding rates against dTTP concentrations fit to *Equation 4* (solid line) to obtain the unwinding $k_{cat}$ and dTTP $K_m$ values. (**D, E**) The $k_{cat}$ and dTTP $K_m$ plotted as a function of GC percentages.
*Figure 2. continued on next page*

*Figure 2. Continued*

(**F**) Schematic of the helicase model with random order of base-capture and dTTP binding. (**G**, **H**) Plots of equilibrium constants for the base-capture steps in the dTTP free ($K_1$) and dTTP bound ($K_2$) state as a function of GC percentages. $K_{d,\ dTTP}$ was fixed at 90 μM, which corresponds to the $K_{m,\ dTTP}$ for the helicase when translocating on ssDNA (*Figure 2—figure supplement 2* and Appendix—Section 4). $k_{cat}$ was fixed at 130 nt/s corresponding to the ssDNA translocation rate of the helicase (*Kim et al., 2002*). Details of the fittings are provided in Appendix—Section 3.

The following figure supplements are available for figure 2:

**Figure supplement 1**. Fitting kinetics of unwinding by T7 helicase.

**Figure supplement 2**. Pi release kinetics.

occur both in the dTTP-free and dTTP-bound states of the leading subunit. We show that the dTTP-bound state is slightly better at capturing the DNA-base than the dTTP-free state (*Figure 2H*), but the equilibrium constants of the base-capture steps in the dTTP-free ($K_1$) and dTTP-bound ($K_2$) states both decrease with increasing GC percentage (*Figure 2G–H*), which indicates that the rate of unwinding by the helicase is limited by base pair separation even at high concentrations of dTTP.

Our results demonstrate that the DNA-unwinding activity of T7 helicase and T7 DNAP is rate-limited by inefficient base-capture at the fork junction. However, the two enzymes respond differently to increasing GC percentage, and this is because of their different kinetic mechanisms. T7 DNAP follows an ordered mechanism, wherein dNTP binding follows the base-capture step. Therefore, the kinetic outcome of increasing GC content is analogous to a pure competitive mechanism where inhibitor (GC content) increases the dNTPs $K_m$ without affecting the unwinding $k_{cat}$. T7 helicase does not follow an ordered mechanism. Consequently, the kinetic outcome of increasing GC content is analogous to a mixed inhibition mechanism where inhibitor (GC content) decreases the unwinding $k_{cat}$ and mildly affects the dTTP $K_m$. Therefore, T7 DNAP is able to achieve fast rates of unwinding at high dNTPs, but this is not the case with the helicase, whose unwinding $k_{cat}$ remains suboptimal even at high-dTTP concentrations.

## Unwinding rates of combined T7 helicase and T7 DNAP enzymes are not rate-limited by base pair separation

We used the same replication fork substrates used in the DNAP experiments to investigate the unwinding mechanism of the combined T7 helicase and T7 DNAP enzymes (*Figure 3A*). In these experiments, helicase and DNAP were preassembled on the fork DNA using dTTP without Mg(II), and unwinding was initiated with Mg(II), dVTPs (3 dNTPs except dTTP), and $dT_{90}$ trap DNA. The kinetic traces show an initial lag followed by an increase in fluorescence, but then a slight dip at the end of the reaction (*Figure 3A*). The dip is more prominent with high-GC content forks than with the low-GC content forks (*Figure 3—figure supplement 1A*). The dip was observed with the isolated helicase and with helicase-DNAP functioning together (*Figure 3—figure supplement 1B*), but not with isolated DNAP. Therefore, we believe that the dip comes from interactions of the helicase with the fluorescein at the end of the lagging strand. We fit the lag and the increase in fluorescence to the $n$-step model to obtain the average base pair unwinding rates. These unwinding rates correlate well with the rates from the gel-based primer-extension assays (*Pandey and Patel, 2014*).

The unwinding rates of the combined enzymes measured at increasing dTTP and constant dVTPs concentration show little dependency on the GC percentage (*Figure 3B*). The unwinding $k_{cat}$ is ~90 bp/s when GC percentage is low, and $k_{cat}$ decreases minimally to 70 bp/s when GC percentage is high. Similarly, the dTTP $K_m$ changes little from 120 to 90 μM as GC percentage increases (*Figure 3C–D*). This indicates that the DNA-unwinding rates of the combined enzymes are not rate-limited by base pair separation at any dNTPs concentrations. Additionally, these results also show that the two enzymes mutually stimulate the activities of unwinding and synthesis.

How does T7 DNAP stimulate the helicase? The dTTP $K_m$ of the isolated helicase on the 50% GC fork is ~160 μM, which decreases to ~80 μM in the presence of T7 DNAP (*Figure 3E*), which is close to the dTTP $K_m$ on ssDNA (*Figure 2—figure supplement 2*). Similarly, the unwinding $k_{cat}$ of the isolated

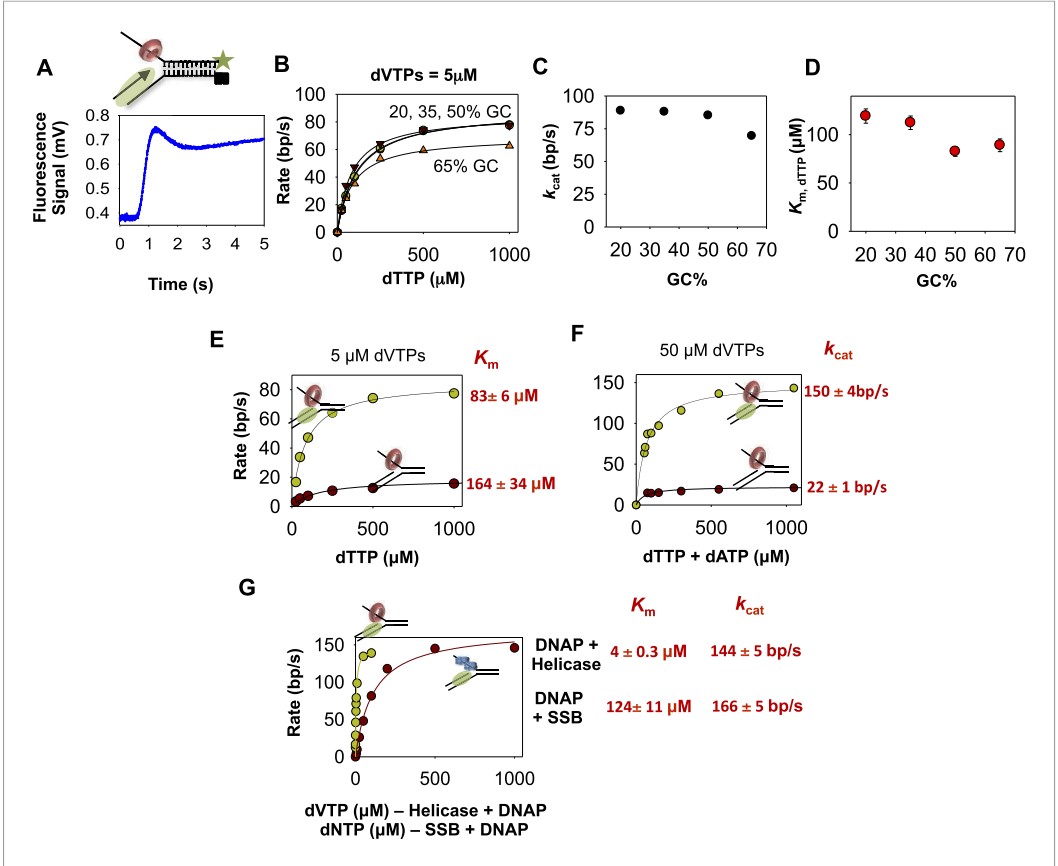

**Figure 3**. The kinetics of DNA unwinding by the combined helicase and DNAP enzymes. (**A**) The replication fork DNA substrate and representative kinetic trace of DNA unwinding by the combined T7 DNAP and T7 helicase enzymes. (**B**) The base pair unwinding rates of the combined enzymes at 5 μM dVTPs plotted against dTTP concentrations and fit to *Equation 4* (solid lines) to obtain the maximal rate of unwinding ($k_{cat}$) and $K_m$ for dTTP. (**C**, **D**) The unwinding $k_{cat}$ and dTTP $K_m$ as a function of GC percentages. (**E**) The unwinding rates of the isolated helicase (red circles) and helicase-DNAP (green circles) were measured using the 50% GC fork at constant 5 μM dVTP and increasing dTTP concentrations. (**F**) The unwinding rates of the isolated helicase (red circle) and helicase-DNAP (green circle) were measured using the 50% GC fork at 50 μM dVTP concentrations and increasing dTTP concentrations. (**G**) The unwinding rates of T7 DNAP with *E. coli* SSB (red circle) or with T7 helicase (green circle) were measured using the 50% GC fork at 500 μM dTTP and increasing dNTPs concentrations.
The following figure supplement is available for figure 3:

**Figure supplement 1**. Unwinding-synthesis trace for helicase-DNAP.

helicase is ∼20 bp/s and increases substantially to ∼150 bp/s in the presence of T7 DNAP (*Figure 3F*), which is close to the translocation rate of the helicase on ssDNA (∼130 nt/s) (*Kim et al., 2002*). Thus, T7 DNAP stimulates T7 helicase by increasing the DNA-unwinding $k_{cat}$ and by slightly decreasing the dTTP $K_m$. The kinetic parameters of T7 helicase coupled to T7 DNAP resemble those of helicase translocating on ssDNA rather than unwinding duplex DNA.

How does the helicase stimulate the DNAP? The unwinding $k_{cat}$ (with SSB) is 160 bp/s and decreases minimally to 140 bp/s in the presence of T7 helicase, whereas the dNTPs $K_m$ decreases by 24-fold from ∼120 μM (with SSB) to 5 μM in the presence of T7 helicase (*Figure 3G*). The 5 μM dNTPs $K_m$ is slightly lower than the DNAP's dNTP $K_m$ on ssDNA template (∼10–20 μM) (*Patel et al., 1991*; *Stano et al., 2005*). Thus, T7 helicase stimulates T7 DNAP by promoting dNTP binding, and in the presence of helicase, T7 DNAP behaves like a motor translocating on ssDNA template.

In summary, by measuring the unwinding kinetics of the individual enzymes and comparing it to the combined enzymes, we determine how T7 helicase and T7 DNAP mutually stimulate each other's

activity. The T7 DNAP stimulates the T7 helicase by increasing the unwinding $k_{cat}$, whereas the helicase stimulates the DNAP by decreasing the dNTPs $K_m$. When functioning independently, the lower $k_{cat}$ of the helicase and higher dNTPs $K_m$ of DNAP are due to inefficient capture of the nucleotide base from the fork junction. This implies that the two enzymes help each other by increasing the efficiency of the base-capture step.

## Coupling and uncoupling of leading strand synthesis

What happens when one enzyme is faster than the other enzyme during leading strand synthesis? Do the enzymes become functionally uncoupled? Does the faster enzyme outrun the slower enzyme? Because T7 helicase mainly uses dTTP as its substrate and the DNAP uses all dNTPs, we can change the concentrations of dTTP and dVTPs to control the speeds of helicase and DNAP, respectively. When the helicase rate is decreased by lowering dTTP, DNA unwinding slows down (*Figure 4A*). At 500 µM dTTP, the $k_{cat}$ of the combined enzymes is fast (150 bp/s), but at 50 µM dTTP, the $k_{cat}$ is 60 bp/s. This implies that when helicase is slower than DNAP, the DNAP will not outrun the helicase. Interestingly, the dVTPs $K_m$ of the combined enzymes remains unchanged at ~5 µM, both at low- and high-dTTP concentrations. This means that when helicase is slower than DNAP, the combined enzymes remain functionally coupled.

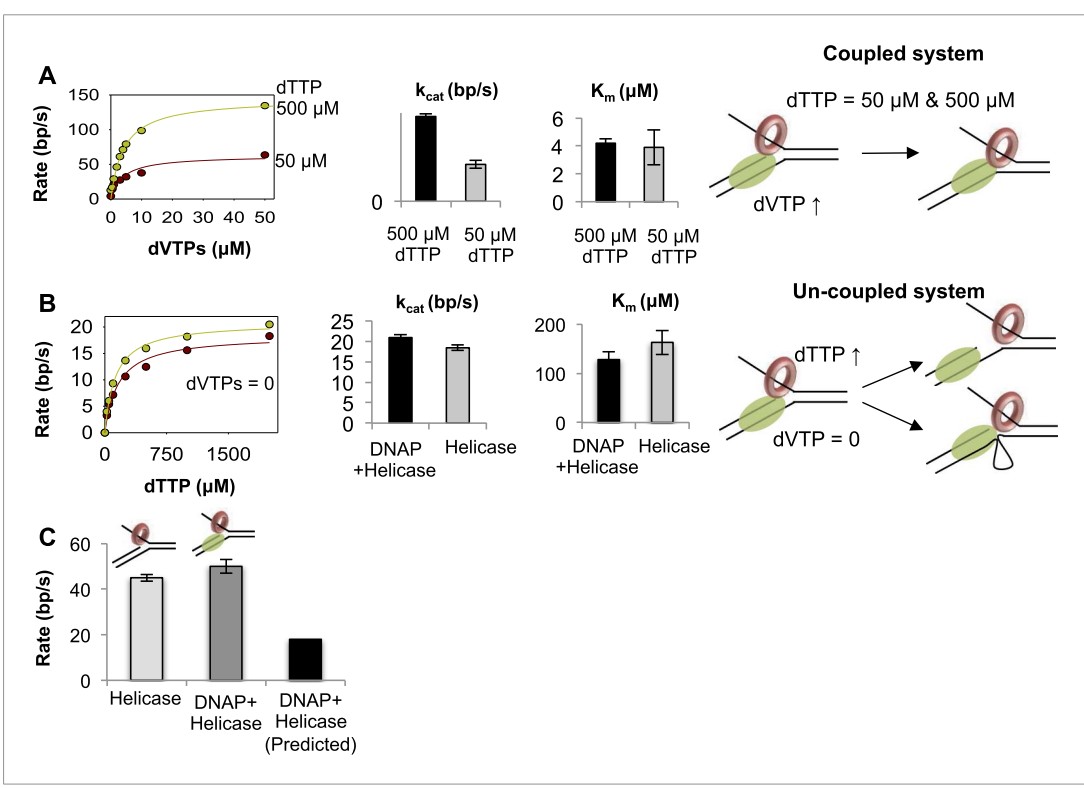

**Figure 4**. Functionally coupled and uncoupled helicase-DNAP. (**A**) The unwinding rates of the combined helicase-DNAP were measured at 50 µM dTTP (red circle) or 500 µM dTTP (green circle) at increasing dVTPs concentration on the 50% GC fork. The bar chart shows the unwinding $k_{cat}$ and dVTPs $K_m$ of the combined enzymes at low- (grey bars) and high-dTTP (black bars) concentrations. The cartoon shows that the enzymes remain functionally coupled when helicase is the slow motor. (**B**) The unwinding rates of the combined enzymes (green circles) at zero dVTPs concentration are compared to the rates of helicase alone (red circles) at increasing dTTP concentrations on the 50% GC fork. The bar charts compare the unwinding $k_{cat}$ and dTTP $K_m$ of helicase-DNAP (black) and isolated helicase (gray). Error bars represent fitting errors. The cartoon show that stalling DNAP leads to functional uncoupling between helicase-DNAP with or without physical uncoupling. (**C**) The DNA unwinding rates of the isolated helicase and helicase-DNAP were measured at 1 mM dTTP and 0.5 µM dVTPs on the 20% GC fork. The bar chart shows the unwinding rate of the isolated helicase (light gray), helicase-DNAP complex (dark gray), and the predicted rate of DNA synthesis by helicase-DNAP assuming coupled synthesis (black).

When DNAP cannot move forward due to lack of dNTPs, the associated helicase unwinds the replication fork with unstimulated rates (*Figure 4B*). This means that the stalled DNAP does not pull the helicase back or prevent it from unwinding the DNA. With stalled DNAP, the unwinding rate of the 50% GC fork is 20 bp/s, which is similar to the rate of helicase functioning independently (18 bp/s). Similarly, with the stalled DNAP, the dTTP $K_m$ of the helicase is ~130 µM, which is similar to the dTTP $K_m$ in the absence of DNAP (~160 µM) (*Figure 4B*). Thus, stalling the DNAP functionally uncouples the helicase. When dVTPs concentrations are low (0.5 µM), the unwinding rate with the combined enzymes is similar to that of the isolated helicase (*Figure 4C*). This indicates that when T7 DNAP is slow, the helicase is capable of moving faster and can outrun the DNAP.

## T7 DNAP unwinds two base pairs and interacts with three nucleotides on the template

The functional data presented above indicate that the two enzymes help each other by increasing the base-capture efficiency. To understand the structural basis for the mutual stimulation of helicase and DNAP, we used 2-AP as a probe to monitor base pair melting. When 2-AP is base-paired (*Figure 5A*), it has a low-fluorescence intensity, but when 2-AP:T base pair is melted and the 2-AP base is unstacked, the fluorescence increases (*Ward et al., 1969*). Such changes in 2-AP fluorescence successfully monitor base unstacking and base pair separation in a variety of enzyme studies, including replication enzymes (*Reha-Krantz, 2009*; *Jose et al., 2012*). However, there are no studies using this method to investigate base pair melting of downstream duplex DNA by replicative DNAPs or the helicase-DNAP complex. By systematically labeling the replication fork DNA with a single 2-AP probe at different positions near the fork junction (*Figure 5B*), we are able to determine the base pair melting footprint of the individual and combined enzymes and deduce the precise positions of the helicase and DNAP at the replication fork.

The crystal structure of T7 DNAP (*Doublie et al., 1998*) shows interactions with two template-bases (N + 1 and N + 2) immediately downstream of the primer-end at positon N and ~90° bend between N + 1 and N + 2 template-bases (*Figure 5C*). Thus, introducing a 2-AP at N + 1 results in a significant increase in fluorescence consistent with unstacking of the N + 1 base from bending of the template DNA (*Figure 5D*). However, there is no increase in fluorescence of 2-AP at N + 2 and a decrease is observed at N + 3 (*Figure 5D*), which is consistent with interactions of these template DNA-bases with the amino acids in the template-binding pocket of T7 DNAP (*Doublie et al., 1998*). The change in 2-AP fluorescence at positions N + 4 and N + 5 is minimal upon DNAP binding, which indicates that T7 DNAP may not interact with these downstream positions. Thus, T7 DNAP influences only three template-bases immediately downstream from the primer-end. Experiments with 2-AP at N + 1 to N + 3 positions on the leading strand show that T7 DNAP makes similar interactions with the template-bases in replication fork substrates (*Figure 5—figure supplement 1D*).

To determine if T7 DNAP unwinds the duplex DNA downstream of the primer-end, we introduced 2-AP at various positions in the lagging strand in the replication fork substrates. The fluorescence intensity increases upon adding T7 DNAP when 2-AP is at the N + 1, N + 2, or N + 3 base pair, but not at N + 4 (*Figure 5E–H*—Green bars). Similarly, when 2-AP is at N + 2 and is part of the fork junction, addition of T7 DNAP increases the fluorescence (*Figure 5—figure supplement 2A*). However, when 2-AP is at N + 3 and is part of the fork junction, no increase in fluorescence is observed (*Figure 5—figure supplement 2B*), which indicates that T7 DNAP does not melt the junction base pair three nucleotides downstream from the primer-end. The increase in 2-AP intensity at N + 3 in the internal position (*Figure 5G*) is due to N + 2 base unstacking and not from unstacking of the N + 3 base. Taken together, these results indicate that T7 DNAP binds to three template-bases downstream of primer-end and melts two base pairs.

When 2-AP experiments with T7 DNAP were carried out with *E. coli* SSB, the fluorescence intensity changes were much larger, although SSB by itself did not increase the 2-AP fluorescence significantly (*Figure 5—figure supplement 3*). These results indicate that T7 DNAP on its own only partially melts the junction base pairs. Furthermore, these base pairs appear to be in dynamic equilibrium (closed ↔ open), because SSB can shift the equilibrium towards the open state by simply binding to ssDNA. This provides direct proof that T7 DNAP can melt the junction base pair, but DNAP is not efficient at preventing base pair reannealing; thus, SSB stimulates the activity of DNAP by trapping the unwound bases.

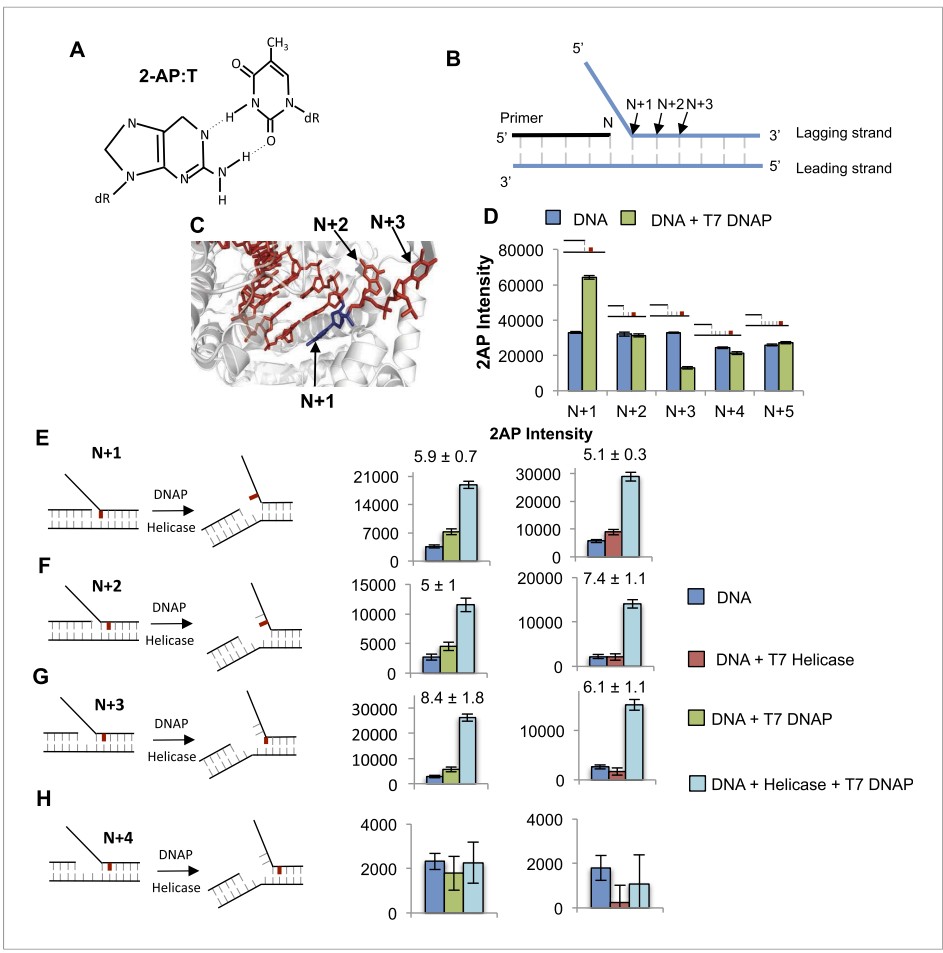

**Figure 5**. Base pair melting by isolated and combined T7 DNAP and T7 helicase using 2-aminopurine fluorescence changes. (**A**) Structure of the 2-aminopurine (2-AP):T base pair. (**B**) Structure of the replication fork substrate for 2-AP studies. The primer-end is N and subsequent base pairs are N + 1, N + 2, etc. The substrates contained a single 2-AP in the leading or the lagging strand. The primer-end is next to the junction base pair as shown or separated by gaps of one to three template strand nucleotides (not shown). (**C**) Crystal structure of T7 DNAP bound to primer-template DNA substrate (PDB: 2AJQ). The N + 1 base (blue) is bound in the insertion site, and the N + 2 and N + 3 bases are bound in the template-binding channel. The figure was made using PyMOL (*Schrodinger, 2010*). (**D**) Fluorescence intensities of 2-AP modified primer-template substrate without (blue) and with T7 DNAP (green). The 2-AP probe is shown in red at the indicated positions. (**E**–**H**) Fluorescence intensities of 2-AP modified replication fork substrates with and without T7 DNAP and T7 helicase. The cartoons show the structure of fork DNA before and after binding of combined T7 DNAP and T7 helicase enzymes. Errors shown are standard deviations from average of 2–5 experiments.

The following figure supplements are available for figure 5:

**Figure supplement 1**. Determining optimal conditions for T7 DNAP and extent of influence of T7 DNAP on template bases in replication fork substrate.

**Figure supplement 2**. Effect of T7 DNAP on 2-AP at the junction at N + 2 and N + 3.

**Figure supplement 3**. Base pair melting by combined SSB—DNAP using 2-AP fluorescence change.

**Figure supplement 4**. Determining optimal conditions for T7 helicase.

**Figure supplement 5**. (**A**–**C**) Fluorescence intensities of fork DNA with 2-AP in the lagging strand at the fork junction and increasing distance between primer end and fork junction with (red bars) and without T7 helicase (blue bars).

## T7 helicase unwinds only the junction base pair and follows the fork junction

The 2-AP experiments with T7 helicase were carried out in the presence of dTMPPCP, the non-hydrolyzable dTTP analog, which does not support translocation or processive DNA unwinding, but is needed for hexamer formation and DNA binding (*Hingorani and Patel, 1993*). T7 helicase does not change the fluorescence intensity of 2-AP in ssDNA (*Figure 5—figure supplement 4B*). However, T7 helicase increases the fluorescence of 2-AP at the fork junction, but not the second base pair from the junction, irrespective of the gap size between the primer end and fork junction (*Figure 5E–H*—Red bars). Thus, unlike T7 DNAP that melts two base pairs upon binding to the replication fork substrate, T7 helicase melts only the junction base pair. However, T7 helicase melts the junction base pair even when the primer-end is separated from the fork junction by more than one nucleotide (*Figure 5—figure supplement 5A–C* compare to *Figure 5—figure supplement 2*). These results indicate that T7 helicase follows the fork junction and is not influenced by the position of the primer-end. Interestingly, SSB has no effect on the helicase catalyzed melting of the fork junction (*Figure 5—figure supplement 5D*). This is consistent with the observation that SSB does not stimulate the unwinding rates of the helicase (*Donmez and Patel, 2008*).

## Synergistic melting of junction base pairs by T7 helicase and T7 DNAP and their precise positions at the fork junction

To investigate the effect of T7 DNAP and T7 helicase together on junction base pair melting, we measured 2-AP fluorescence after sequential addition of helicase and DNAP to the fork DNA in that order and the reverse order (*Figure 5E–H*—light blue bars). The combined enzymes show a much larger increase in fluorescence intensity with 2-AP at N + 1, N + 2, or N + 3 base pairs, but not at N + 4. As shown earlier with the DNAP, the increase in N + 3 is due to N + 2 base unstacking; therefore, the results indicate that the combined enzymes melt two base pairs downstream from the primer-end, just like T7 DNAP, but more efficiently.

The 2-AP fluorescence intensity at steady state measures the equilibrium distribution of melted and annealed states of the junction base-pair. The small increase in fluorescence intensity with the isolated helicase and DNAP suggests that each enzyme shifts the equilibrium only moderately to the base-pair melted state. On the other hand, the striking increase in fluorescence intensity with the combined enzymes indicates that together the two enzymes shift the equilibrium strongly toward the base-pair melted state. Interestingly, the combined effect of helicase and DNAP on base pairs melting is greater than the sum, which indicates synergism in DNA melting. This synergism depends on the number of nucleotides between the primer-end and fork junction. Synergistic melting of the base pair is observed only when there is no gap or one nucleotide gap between the primer-end (DNAP-binding site) and fork junction (helicase-binding site) (*Figure 5E–G*). Synergistic melting is not observed when there are two nucleotides between the primer-end and fork junction (*Figure 5H*). The results also demonstrate that a replication fork with two ssDNA template-bases between the primer-end and fork junction can stably accommodate both enzymes of the T7 replisome. Therefore, this study defines the specific positions of helicase and DNAP at the replication fork junction with single-base resolution to create a structural model of the replisome (*Figure 6*) that forms the basis for understanding how the helicase and DNAP mutually stimulate each other's activities as discussed below.

## Discussion

The studies here use 2-AP fluorescence changes and provide direct evidence that both T7 DNAP and T7 helicase can melt the junction base pair. In fact, T7 DNAP by itself melts two base pairs upon binding to the replication fork DNA. This was unexpected, because DNA synthesis occurs in steps of one nucleotide, which would require unwinding of only one base pair at a time. Based on our 2-AP studies here and the crystal structure of T7 DNAP (*Doublie et al., 1998*), we propose that T7 DNAP destabilizes two base pairs of duplex DNA upon initial binding to the forked DNA to create two ss nucleotides for its template-binding pocket. Two or more template-bases in the downstream template-binding pocket have been observed in crystal structures of other DNAPs as well (*Doublie et al., 1998*; *Johnson et al., 2003*; *Berman et al., 2007*; *Jain et al., 2014*). Hence, this mode of DNA binding could be a general feature of replicative DNAPs. We show that T7 helicase on its own can also melt the junction base pair, but it melts only one base pair and follows the fork junction.

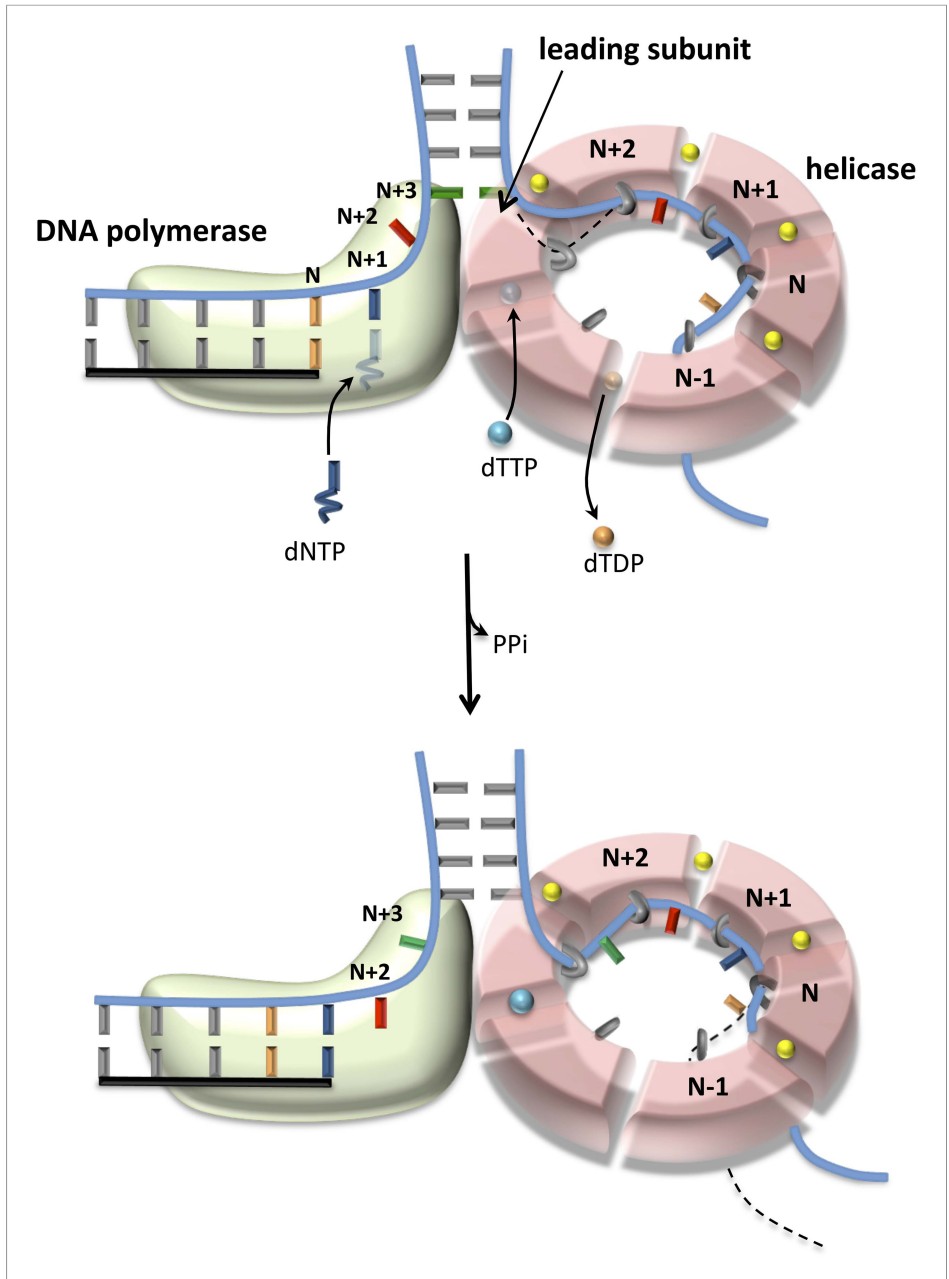

**Figure 6**. Proposed model of DNA unwinding-synthesis by T7 replisome. The top cartoon of T7 replisome results after melting of the N + 2 base pair by helicase and DNAP. There are two unwound nucleotides (N + 1 and N + 2) between the primer-end and fork junction at N + 3. The N + 1 base of the leading strand is bound in the insertion site and serves as the templating nucleotide for the incoming dNTP, and the N + 2 is unstacked and bound in the template-binding channel of the DNAP. The complementary N + 1 and N + 2 nucleotides (as well as N and N − 1) on the lagging strand are bound to individual subunits of the helicase hexamer, as shown. Stable binding to N + 2 base by the helicase triggers dTTP hydrolysis and products release at different subunits of the ring. The helicase subunit at the leading edge has weak interactions with the partially unwound N + 3 junction base (dotted line), which gets stabilized after the next round of catalysis. When N is elongated by one nucleotide, the N + 2 moves into the insertion site after PPi release, and the helicase and DNAP cooperatively melt the N + 3 junction base pair, as shown in the bottom cartoon. This model explains the one-nucleotide step size where the combined enzymes translocate by one nucleotide for every dTTP hydrolyzed and nucleotide incorporated (**Pandey and Patel, 2014**).

Comparing DNA melting by isolated and combined enzymes reveals that the junction base pair is melted only partially by the isolated enzymes (*Figure 5E–H*). This explains the slow and GC sensitive unwinding rates of the isolated enzymes in our kinetic experiments (*Figures 1C, 2C*). The junction base pair is melted more efficiently by the combined enzymes, explaining fast and GC-independent rates of the combined enzymes (*Figure 3B*). We observed that the combined enzymes melted two base pairs upon binding to the fork DNA. We propose that this occurs *initially* to establish the catalytically competent structure of the replication fork—T7 DNAP melts two base pairs and binds two ssDNA nucleotides in its template-binding pocket, which positions the helicase two nucleotides ahead at the fork junction, as shown in *Figure 6*. We propose that during active leading strand synthesis, the enzymes melt only one base pair at the fork junction at a time consistent with the one nucleotide chemical step size of the combined enzymes (*Pandey and Patel, 2014*).

## Model of the T7 replisome

When DNAP is in the post-translocated state, the N + 1 templating base (blue) is positioned in the polymerase active site ready to base pair with the incoming dNTP (*Figure 6*, upper cartoon). After the chemical step, the DNAP translocates downstream by one nucleotide to position the next templating-base N + 2 (red) in the active site (*Figure 6*, lower cartoon). This forward translocation step is coupled to unwinding of the N + 3 junction base pair (green). Based on our results, we propose that the DNAP by itself is not efficient at preventing junction base pair reannealing, and this unfavorable equilibrium constant for DNA melting destabilizes the post-translocated state of DNAP and competes with incoming dNTP binding. When helicase is present at the fork junction, it helps both unwind and trap the junction bases. Thus, the associated helicase stimulates dNTP binding by stabilizing the post-translocated state of T7 DNAP, and the outcome is decrease in dNTPs $K_m$. The helicase by itself is not efficient at unwinding the fork junction. However, the associated DNAP by providing an unwound base to the helicase at the fork junction facilitates the base-capture step and drives the reactions of dTTP binding-hydrolysis-product release around the helicase ring, and the outcome is an increase in the unwinding $k_{cat}$. The combined binding energy of the two enzymes bound to opposite strands is sufficient to keep the unwound bases from reannealing, explaining the fast and GC-independent unwinding rates of helicase-DNAP.

Interestingly, cooperative and enhanced efficiency of base pair melting is observed only when the helicase and DNAP are within one nucleotide distance from each other. In most cases, helicase is coupled physically to the DNAP, either directly as in the case of T7 replication system or indirectly through accessory proteins (*Kim et al., 1996*; *Hamdan et al., 2007*; *Gambus et al., 2009*; *Sengupta et al., 2013*). Some of these interactions aid in the assembly of the replisome (*Zhang et al., 2011*) and perhaps in proper positioning of the helicase with the DNAP in the replisome, but the consequences of breaking physical interactions on synergistic melting need to be investigated.

One can imagine situations where flexible positioning is needed when one or the other enzyme pauses or stalls during leading strand synthesis. Our investigation of such situations reveals that when DNAP stalls or is the slower motor, the helicase becomes functionally uncoupled and outruns the DNAP by unwinding the replication fork at the unstimulated rates. Similar behaviors were observed in other replisome studies as well (*Byun et al., 2005*; *McInerney and O'Donnell, 2007*). Whether the functionally uncoupled helicase remains physically coupled to the DNAP remains unknown. Interestingly, when the helicase slows down, the two enzymes remain functionally coupled as evident from the low dNTPs $K_m$ and that the DNAP does not outrun the helicase. In this case, the combined enzymes unwind the DNA with the stimulated rate of the helicase.

Although SSB stimulates base pair melting by T7 DNAP, our studies find that the unwinding rates of T7 DNAP with SSB remain GC-sensitive at low-dNTPs concentrations. We propose that this is because SSB cannot trap the junction bases coordinately with DNA synthesis in the manner that T7 helicase does during leading strand synthesis. Similarly, it has been shown previously that SSB does not increase the unwinding rates of T7 helicase, which remain GC sensitive at all concentrations of dTTP (*Donmez and Patel, 2008*). These observations indicate that simply trapping the displaced strand by DNA binding is not sufficient, but coordination between the steps of junction base pair unwinding/trapping and synthesis is needed for rate acceleration. The replicative helicase is a central player in coordinating leading and lagging strand synthesis (*Pandey et al., 2009*). The interdependency between helicase and DNAP assures that the DNA is not unwound in an uncoupled manner leading to disruption in the coordinated synthesis of the two strands.

The mechanism of DNAP is conserved in all organisms where DNAPs elongate the primer in the 3′–5′ direction. On the other hand, replicative helicases of the prokaryotes and phages unwind DNA in the 5′–3′ direction, whereas those of eukaryotes unwind DNA in the opposite 3′–5′ direction. Our studies suggest that the leading edges of the T7 helicase and T7 DNAP are close together at the fork junction and this conformation is important for functional coupling of unwinding and synthesis reactions and preventing DNA reannealing. This model of the replication fork is likely to be generally applicable to replisomes of prokaryotes as most show functional coupling between helicase and DNAP (*Patel et al., 2011*). In contrast to prokaryotic replisomes, the replicative helicase of eukaryotes and archaea binds to the same strand as the DNAP (*O'Donnell et al., 2013*). In this case, both helicase and DNAP cannot be close to the fork junction, and there must be other mechanisms to functionally couple the two activities and prevent junction base pair reannealing. It is possible that although the MCM2-7 helicase encircles the leading strand, other subunits in the CMG (Cdc45-MCM2-7-GINS) complex may interact with the lagging strand and this could be a mechanism for preventing DNA reannealing at the fork junction (*Costa et al., 2014*).

# Materials and methods

## Oligonucleotides and proteins

Oligodeoxynucleotides labeled with fluorescein (FAM) on the 3′-end and BHQ1 on the 5′-end were purchased from Biosearch Technologies and RP-HPLC purified (Novato, CA). Oligodeoxynucleotides labeled with fluorescein on the 5′-end, 2-AP labeled and unlabeled oligodeoxynucleotides were purchased from Integrated DNA Technologies (Coralville, IA). These DNAs were gel-purified and extracted from the gel by electroelution (Whatman Schleicher & Schuell). Replication fork substrates were created by heating the appropriate DNA strands to 95°C for 5 min and slowly cooling to room temperature. The DNA sequences are provided in *Supplementary file 1*.

T7 helicase (gp4A′), T7 gp5 exo-, and *E. coli* SSB were purified as described (*Lohman et al., 1986*; *Patel et al., 1991*, *1992*; *Kim et al., 1992*). Thioredoxin was purchased from Sigma (St. Louis, MO).

## Real-time DNA-unwinding assays

The unwinding assays were carried out at 18°C in a stopped-flow instrument (Kintek Corp, Austin, TX) with excitation at 480 nm and fluorescence emission using a long pass 515 nm cut-off filter. Reaction buffer A for the helicase and helicase-DNAP experiments contained 50 mM Tris acetate, pH 7.6, 50 mM potassium glutamate, 1.5 mM Ethylenediaminetetraacetic acid (EDTA), 5 mM Dithiothreitol (DTT), 10 mM total Mg(II). For the helicase assays, a mixture of fork DNA, T7 helicase, dTTP, and EDTA from syringe A was mixed with Mg(II), dVTPs (dCTP, dGTP, and dATP), and $dT_{90}$ trap from syringe B to initiate the reaction. The reactions with the combined enzymes were carried out similarly except syringe A contained T7 DNAP. For the T7 DNAP-unwinding assays, a mixture of fork DNA, T7 DNAP, and *E. coli* SSB (pre-incubated at 18°C for 10 min) from syringe A was mixed with Mg(II) and dNTPs from syringe B to initiate the reaction. Reaction buffer B for the DNAP reactions contained 50 mM Tris Cl, pH 7.6, 40 mM NaCl, 1.5 mM EDTA, 5 mM DTT, 8.1–8.5 mM free Mg(II) ($MgCl_2$). Reaction buffer B was used in the T7 DNAP-unwinding assays as SSB was observed to precipitate in buffer A. The final concentrations of enzymes and DNA were 10 nM fork DNA, 20 nM T7 helicase hexamer, 20 nM T7 DNAP, 200 nM *E. coli* SSB, and 2 µM $dT_{90}$ trap DNA.

## 2-AP fluorescence studies

The equilibrium fluorescence experiments were carried out on FluoroMax 4 (Horiba Join Yvon Inc). The sample was excited at 315 nm (2 mm slit width), and emission was measured at 370 nm (6 mm slit width). The buffer contained 50 mM Tris Cl pH 7.6, 40 mM NaCl, 10 mM $MgCl_2$, 5 mM DTT. The observed fluorescence was corrected for buffer and protein bound to unlabeled replication fork substrate. The proteins absorb minimally at the emission wavelength, and hence, the inner filter effect was negligible. The experiments were carried out with 100 nM DNA, 200 nM T7 helicase (*Figure 5—figure supplements 1B, 2A*), 200 nM T7 DNAP/thioredoxin (2.5 times excess thioredoxin) (*Figure 5—figure supplement 1A*), and 10 µM dTMP-PCP at 25°C. A sample trace showing the effect of DNAP and helicase binding to a 2-AP DNA substrate is shown in *Figure 5—figure supplements 1C, 2C*.

## Kinetic data analysis

The DNA unwinding kinetics were fit to the *n*-step model (*Ali and Lohman, 1997*) using *gfit* and model [unwinding.m] in MATLAB with Optimization toolbox (The MathWorks, Inc., Natick, MA) (*Levin et al., 2009*). Unwinding is modeled as a multistep process with equal step-size (*s*) and rate constant ($k_i$) that are estimated from fittings as described previously (*Pandey et al., 2010*). More information about the fitting is provided in the Appendix—Methods section. The average unwinding rates were plotted against dNTP concentration and fit to the hyperbolic equation to obtain $k_{cat}$ and $K_m$ values.

$$unwinding\ rate = \frac{k_{cat}*[dNTP]}{K_m + [dNTP]}. \tag{1}$$

## Acknowledgements

We would like to thank Dr Sanjay Tyagi for suggestions on suitable fluorophore combinations for designing the unwinding-synthesis DNA substrates. We would like to thank Gayatri Patel for helping in purifying the T7 helicase enzyme. We would also like to thank Dr Yong Joo Jeong for determining the $K_{1/2,\ dTTP}$ in the presence of $dT_{90}$ ssDNA. This work was supported by the NIH grant GM55310 to SSP.

## Additional information

### Funding

| Funder | Grant reference | Author |
| --- | --- | --- |
| National Institutes of Health (NIH) | GM55310 | Smita S Patel |

The funder had no role in study design, data collection and interpretation, or the decision to submit the work for publication.

### Author contributions

DN, Conception and design, Acquisition of data, Analysis and interpretation of data, Drafting or revising the article; MP, Drafting or revising the article, Contributed unpublished essential data or reagents; SSP, Conception and design, Analysis and interpretation of data, Drafting or revising the article

## Additional files

### Supplementary file

• Supplementary file 1. DNA sequences used in unwinding and 2-aminopurine experiments.

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

# Appendix

## Section 1: The kinetics of DNA unwinding by T7 DNAP in the presence of *E. coli* SSB

The DNA unwinding kinetics show an initial time lag followed by an increase in fluorescence in two phases, a fast phase and a slow phase (**Figure 1—figure supplement 2A**). We confirmed that the slow phase does not result from SSB binding to released ssDNA, because ssDNA + SSB does not lead to an increase in fluorescence of ssDNA with fluorescein at the 3′ end (**Figure 1—figure supplement 2B**). The slow phase could arise from a population of T7 DNAP that unwinds the fork DNA at an overall slower rate perhaps due to frequent stalling or a population not assisted by SSB. Increasing *E. coli* SSB concentration resulted in the reduction of the slow phase amplitude (**Figure 1—figure supplement 2C**). Therefore, the unwinding rates by T7 DNAP were estimated by fitting the fast phase to the stepping model. The fast phase rate matches closely with the strand displacement rate obtained from the gel-based assays (**Pandey and Patel, 2014**).

## Section 2: Modeling the kinetics of DNA unwinding by T7 DNAP with *E. coli* SSB

The three-step model of T7 DNAP:

$$K_1 = [Pol_n \cdot Base]/[Pol_n]; K_2 = [Pol_n \cdot Base][dNTP]/[Pol_n \cdot Base \cdot dNTP],$$

$$\text{Rate of DNA unwinding} = k_{cat}*[Pol_n \cdot Base \cdot dNTP],$$

$$\text{Enzyme total} = [Pol_n] + [Pol_n \cdot Base] + [Pol_n \cdot Base \cdot dNTP]$$
$$= [Pol_n \cdot Base]/K_1 + [Pol_n \cdot Base] + [Pol_n \cdot Base][dNTP]/K_2,$$

$$\frac{\textbf{Rate of DNA unwinding}}{\textbf{Enzyme total}} = \frac{k_{cat}*[\textbf{Pol}_n \cdot \textbf{Base} \cdot \textbf{dNTP}]}{[\textbf{Pol}_n \cdot \textbf{Base}]\left(1 + \frac{1}{K_1} + \frac{[\textbf{dNTP}]}{K_2}\right)},$$

$$= \frac{k_{cat}*\frac{[\textbf{Pol}_n \cdot \textbf{Base}][\textbf{dNTP}]}{K_2}}{[\textbf{Pol}_n \cdot \textbf{Base}]\left(1 + \frac{1}{K_1} + \frac{[\textbf{dNTP}]}{K_2}\right)},$$

$$= \frac{k_{cat}*[\textbf{dNTP}]/K_2}{(K_2 K_1 + K_2 + K_1[\textbf{dNTP}])/K_2},$$

$$\textbf{Rate of DNA unwinding} = \frac{k_{cat}*[\textbf{dNTP}]}{\left(K_2\left(1 + \frac{1}{K_1}\right) + [\textbf{dNTP}]\right)},$$

$$\text{Observed } K_m = K_2*(1 + 1/K_1),$$

where $K_2$ is the $K_d$ for dNTP and fixed at 10 µM based on literature data for the T7 DNAP (**Patel et al., 1991**; **Stano et al., 2005**). $k_{cat}$ was allowed to float but made a shared parameter.

## Section 3: Modeling the kinetics of DNA unwinding by T7 helicase
### Model A

$$[Helicase_n] \underset{K_1}{\overset{Base}{\rightleftharpoons}} [Helicase_n \bullet base] \underset{K_{d, dTTP}}{\overset{dTTP}{\rightleftharpoons}} [Helicase_n \bullet base \bullet dTTP] \xrightarrow{dTDP + Pi} [Helicase_{n+1}]$$

$$K_1 = [\text{Helicase} \cdot \text{Base}]/[\text{Helicase}],$$

$$K_{d, dTTP} = [\text{Helicase} \cdot \text{Base}][\text{dTTP}]/[\text{Helicase} \cdot \text{Base} \cdot \text{dTTP}],$$

$$\text{Rate of DNA unwinding} = k_{cat}*[\text{Helicase} \cdot \text{Base} \cdot \text{dTTP}],$$

$$\text{Enzyme total} = [\text{Helicase}] + [\text{Helicase} \cdot \text{Base}] + [\text{Helicase} \cdot \text{Base} \cdot \text{dTTP}]$$
$$= [\text{Helicase} \cdot \text{Base}]/K_1 + [\text{Helicase} \cdot \text{Base}] + [\text{Helicase} \cdot \text{Base}][\text{dTTP}]/K_{d, dTTP},$$

$$\frac{\textbf{Rate of DNA unwinding}}{\textbf{Enzyme total}} = \frac{k_{cat}*[\textbf{Helicase} \cdot \textbf{Base} \cdot \textbf{dTTP}]}{[\textbf{Helicase} \cdot \textbf{Base}]\left(1 + \frac{1}{K_1} + \frac{[\textbf{dTTP}]}{K_{d,dTTP}}\right)},$$

$$= \frac{k_{cat}* \frac{[\textbf{Helicase} \cdot \textbf{Base}][\textbf{dTTP}]}{K_{d,dTTP}}}{[\textbf{Helicase} \cdot \textbf{Base}]\left(1 + \frac{1}{K_1} + \frac{[\textbf{dTTP}]}{K_{d,dTTP}}\right)},$$

$$= \frac{k_{cat}*[\textbf{dTTP}]}{\left(K_{d,dTTP}*\left(1 + \frac{1}{K_1}\right) + [\textbf{dTTP}]\right)},$$

$$\textbf{Rate of DNA unwinding} = \frac{k_{cat}*[\textbf{dTTP}]}{K_{d,dTTP}*\left(1 + \frac{1}{K_1}\right) + [\textbf{dTTP}]},$$

$$\text{Observed } K_m = K_{d, dTTP}*(1 + 1/K_1); \text{ Observed } k_{cat} = k_{cat},$$

where $K_{d, dTTP}$ is the dTTP $K_d$ of the leading subunit of the helicase when bound to ssDNA base, which was fixed at 90 μM based on dTTP $K_m$ of T7 helicase in the presence of dT$_{90}$ DNA (**Figure 2—figure supplement 2** and Appendix—Section 4). The $k_{cat}$ was fixed at 130 nt/s, which corresponds to the rate of translocation by the helicase on ssDNA (**Kim et al., 2002**). The $K_1$ is the equilibrium constant for the base-capture step. This model predicts that $k_{cat}$ remains constant with increasing GC content, which is not what we observed (**Figure 2D**).

### Model B

$$[Helicase_n] \underset{K_{d*, dTTP}}{\overset{dTTP}{\rightleftharpoons}} [Helicase_n \bullet dTTP] \underset{K_1}{\overset{Base}{\rightleftharpoons}} [Helicase_n \bullet base \bullet dTTP] \xrightarrow{dTDP + Pi} [Helicase_{n+1}]$$

$$K_{d*, dTTP} = [\text{Helicase}][\text{dTTP}]/[\text{Helicase} \cdot \text{dTTP}],$$

$$K_2 = [\text{Helicase} \cdot \text{Base} \cdot \text{dTTP}]/[\text{Helicase} \cdot \text{dTTP}],$$

$$\text{Rate of DNA unwinding} = k_{cat}{}^*[\text{Helicase} \cdot \text{Base} \cdot \text{dTTP}],$$

$$
\begin{aligned}
\text{Enzyme total} &= [\text{Helicase}] + [\text{Helicase} \cdot \text{dTTP}] + [\text{Helicase} \cdot \text{Base} \cdot \text{dTTP}] \\
&= [\text{Helicase}] + [\text{Helicase}][\text{dTTP}]/K_{d^*, \text{dTTP}} + K_2[\text{Helicase} \cdot \text{dTTP}] \\
&= [\text{Helicase}] + [\text{Helicase}][\text{dTTP}]/K_{d^*, \text{dTTP}} + K_2[\text{Helicase}][\text{dTTP}]/K_{d^*, \text{dTTP}},
\end{aligned}
$$

$$
\frac{\textbf{\textit{Rate of DNA unwinding}}}{\textbf{\textit{Enzyme total}}} = \frac{k_{cat}{}^*[\textbf{\textit{Helicase}} \cdot \textbf{\textit{Base}} \cdot \textbf{\textit{dTTP}}]}{[\textbf{\textit{Helicase}}]\left(1 + \frac{K_2[\textbf{\textit{dTTP}}]}{K_{d^*\,\text{dTTP}}} + \frac{[\textbf{\textit{dTTP}}]}{K_{d^*\,\text{dTTP}}}\right)},
$$

$$
= \frac{k_{cat}{}^*K_2[\textbf{\textit{Helicase}} \cdot \textbf{\textit{dTTP}}]}{[\textbf{\textit{Helicase}}]\left(1 + \frac{(1+K_2)[\textbf{\textit{dTTP}}]}{K_{d^*\,\text{dTTP}}}\right)},
$$

$$
= \frac{k_{cat}{}^* \frac{K_2[\textbf{\textit{Helicase}}][\textbf{\textit{dTTP}}]}{K_{d^*\,\text{dTTP}}}}{[\textbf{\textit{Helicase}}]\frac{(K_{d^*\,\text{dTTP}} + (1+K_2)[\textbf{\textit{dTTP}}])}{K_{d^*\,\text{dTTP}}}},
$$

$$
= \frac{k_{cat}{}^* \frac{K_2[\textbf{\textit{dTTP}}]}{(1+K_2)}}{\frac{K_{d^*\,\text{dTTP}}}{(1+K_2)} + [\textbf{\textit{dTTP}}]},
$$

$$
\textbf{\textit{Rate of DNA unwinding}} = \frac{k_{cat}{}^* \frac{K_2[\textbf{\textit{dTTP}}]}{(1+K_2)}}{\frac{K_{d^*\,\text{dTTP}}}{(1+K_2)} + [\textbf{\textit{dTTP}}]},
$$

Observe $K_m = K_{d^*, \text{dTTP}}/(1 + K_2)$; Observed $k_{cat} = k_{cat}{}^*K_2/(1 + K_2)$,

where $K_2$ is the equilibrium constant for the Helicase dTTP to Helicase Base dTTP step. $K_{d^*, \text{dTTP}}$ is the $K_d$ for dTTP when the leading subunit is not bound to the DNA base. As this value is unknown, $K_{d^*, \text{dTTP}}$ was allowed to float but made a global parameter for the different GC DNA substrates. $k_{cat}$ was fixed at 130 nt/s, which corresponds to the rate of translocation by the helicase on ssDNA (**Kim et al., 2002**). Although this model fits well for the 20% and 35% GC DNA substrates and predicted a weak dTTP affinity ($K_{d^*, \text{dTTP}}$ = 530 µM) for the leading subunit in the absence of DNA (**Figure 2—figure supplement 1B**), the dTTP dependency data showed a poor fit for the 50% and 65% GC DNA substrates (**Figure 2—figure supplement 1C**).

## Model C

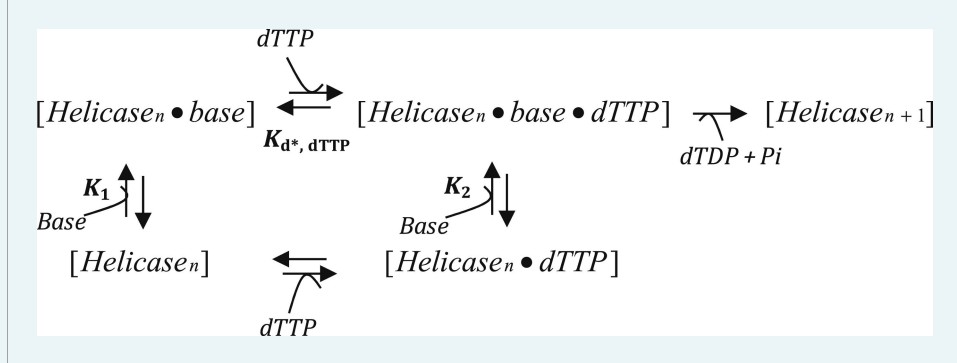

$$K_1 = [\text{Helicase} \cdot \text{Base}]/[\text{Helicase}],$$

$$K_2 = [\text{Helicase} \cdot \text{Base} \cdot \text{dTTP}]/[\text{Helicase} \cdot \text{dTTP}],$$

$$K_{d, \text{dTTP}} = [\text{Helicase} \cdot \text{Base}][\text{dTTP}]/[\text{Helicase} \cdot \text{Base} \cdot \text{dTTP}],$$

$$\text{Rate of DNA synthesis} = k_{cat}*[\text{Helicase} \cdot \text{Base} \cdot \text{dTTP}],$$

$$
\begin{aligned}
\text{Enzyme total} \quad &= [\text{Helicase}] + [\text{Helicase} \cdot \text{Base}] + [\text{Helicase} \cdot \text{dTTP}] + [\text{Helicase} \cdot \text{Base} \cdot \text{dTTP}] \\
&= [\text{Helicase} \cdot \text{Base}]/K_1 + [\text{Helicase} \cdot \text{Base}] + [\text{Helicase} \cdot \text{Base} \cdot \text{dTTP}]/ \\
&\quad K_2 + [\text{Helicase} \cdot \text{Base} \cdot \text{dTTP}] \\
&= [\text{Helicase} \cdot \text{Base}]/K_1 + [\text{Helicase} \cdot \text{Base}] + [\text{Helicase} \cdot \text{Base}][\text{dTTP}]/ \\
&\quad K_2*K_{d,\text{dTTP}} + [\text{Helicase} \cdot \text{Base}][\text{dTTP}]/K_{d, \text{dTTP}},
\end{aligned}
$$

$$\frac{\textbf{Rate of DNA unwinding}}{\textbf{Enzyme total}} = \frac{k_{\textbf{cat}}*[\textbf{Helicase} \cdot \textbf{Base} \cdot \textbf{dTTP}]}{[\textbf{Helicase} \cdot \textbf{Base}]\left(1 + \frac{1}{K_1} + \frac{[\textbf{dTTP}]}{K_2* K_{d,\textbf{dTTP}}} + \frac{[\textbf{dTTP}]}{K_{d,\textbf{dTTP}}}\right)},$$

$$= \frac{k_{\textbf{cat}}*\frac{[\textbf{Helicase} \cdot \textbf{Base}][\textbf{dTTP}]}{K_{d,\textbf{dTTP}}}}{[\textbf{Helicase} \cdot \textbf{Base}]\left(1 + \frac{1}{K_1} + [\textbf{dTTP}]\left(\frac{(1+K_2)}{K_2* K_{d,\textbf{dTTP}}}\right)\right)},$$

$$= \frac{k_{\textbf{cat}}*\frac{[\textbf{dTTP}]}{K_{d,\textbf{dTTP}}}}{\frac{\left(K_2* K_{d,\textbf{dTTP}}(1+K_1) + [\textbf{dTTP}]*K_1*(1+K_2)\right)}{K_1* K_2* K_{d,\textbf{dTTP}}}},$$

$$= \frac{k_{\textbf{cat}}*\frac{[\textbf{dTTP}]}{K_{d,\textbf{dTTP}}}}{\frac{K_{d,\textbf{dTTP}} * K_2 * (1+K_1)}{K_1* K_2} + \frac{K_1*(1+K_2)*[\textbf{dTTP}]}{K_1* K_2}},$$

$$\textbf{Rate of DNA unwinding} = \frac{k_{\textbf{cat}}*\frac{[\textbf{dTTP}]* K_2}{(1+K_2)}}{\frac{K_{d,\textbf{dTTP}}(1+ K_1)* K_2}{(1+ K_2)*K_1} + [\textbf{dTTP}]},$$

$$\text{Observed } K_m = K_{d, \text{dTTP}}*K_2(1+K_1)/K_1(1 + K_2); \text{ Observed } k_{cat} = k_{cat}*K_2/(1 + K_2),$$

where $K_1$ is the equilibrium constant for the Helicase to the Helicase dTTP state, $K_2$ is the equilibrium constant for the Helicase Base to the Helicase Base dTTP state. $K_{d, \text{dTTP}}$ is the $K_d$ for dTTP when the helicase is bound to DNA base and is fixed at 90 µM based on dTTP $K_m$ of T7 helicase in the presence of $dT_{90}$ DNA (*Figure 2—figure supplement 2*). $k_{cat}$ was fixed at 130 nt/s, which corresponds to the rate of translocation by the helicase on ssDNA (*Kim et al., 2002*).

## Section 4: The $K_{d, \text{dTTP}}$ of the leading helicase subunit bound to DNA

It is difficult to directly measure the apparent $K_{d, \text{dTTP}}$ of the leading subunit of the hexameric helicase that has captured the DNA base at the fork junction. We reasoned that the dTTP $K_m$ of T7 helicase translocating on ssDNA would be the closest estimate of the dTTP $K_d$ of the base-captured state of the leading subunit, because the base-capture and dTTP binding steps on ssDNA are not limited by base pair separation. The dTTP $K_m$ of 90 µM was measured using $dT_{90}$ ssDNA (that lacks any secondary structure) and pre-steady state Pi release rates as a function of dTTP concentrations (*Figure 2—figure supplement 2*). Note that the 90 µM is a composite dTTP $K_d$ that includes the steps of dTTP binding, hydrolysis, and Pi release, which is adequate for our model (Section 3) that combines these steps into one chemical step.

## Section 5: Appendix methods
### Pre-steady state kinetics of dTTP hydrolysis
The pre-steady state kinetics of dTTP hydrolysis was measured as described earlier (*Kim et al., 2002*) in the presence of $dT_{90}$ ssDNA at increasing concentration of dTTP,

and the dTTP dependence was fit to a hyperbola to obtain the $K_{1/2,\ dTTP}$ in the presence of ssDNA.

## Kinetic data analysis

The DNA-unwinding kinetics were fit to the $n$-step model (**Ali and Lohman, 1997**) using gfit and model [unwinding.m] in MATLAB with Optimization toolbox (The MathWorks, Inc., Natick, MA) (**Levin et al., 2009**). Unwinding is modeled as a multistep process with equal step-size ($s$) and rate constant ($k_i$) that are estimated from fittings as described previously (**Pandey et al., 2010**). To account for heterogeneity, the model calculates the sum of $N$ unwinding processes. The stepping kinetics are described by the following equations.

$$F(t) = \sum_i^N A_i \Gamma(n_i, k_i, t) + F_o,\qquad(2)$$

where,

$$\Gamma(n_i, k_i, t) = \frac{\int_0^{k_i t} e^{-x} x^{n-1} dx}{\int_0^\infty e^{-x} x^{n-1} dx}.\qquad(3)$$

$F$ is the fraction of DNA substrates molecules that have completely undergone strand displacement synthesis, $A_i$ is the amplitude and $t$ is reaction time. The number of steps, $n$, is given by

$$n = \frac{L - L_m}{s},\qquad(4)$$

where, $L$ is the number of base pairs in the dsDNA to be unwound, $L_m$ is the length of the shortest dsDNA that can stay together between reaction termination and data observation. $L_m$ is negligible under stopped flow assay conditions where there is no lag between reaction termination and data observation (**Donmez and Patel, 2008**) and hence set to 0. The average base pair unwinding rate is $k_i \times s$. Best fits were obtained assuming unwinding by more than one population with identical step size, but different stepping rates.

