## [Decision Letter]

Thank you for sending your work entitled “Cooperative base pair melting by helicase and polymerase positioned one nucleotide from each other” for consideration at *eLife*. Your article has been favorably evaluated by John Kuriyan (Senior editor) and three reviewers, one of whom is a member of our Board of Reviewing Editors.

The Reviewing editor and the other reviewers discussed their comments before we reached this decision, and the Reviewing editor has assembled the following comments to help you prepare a revised submission.

This manuscript furthers our understanding of the functional coupling between helicase and DNA polymerase for rapid and efficient DNA replication. The authors varied the cofactor concentration in a series of unwinding assays to determine kinetic parameters highlighting the stimulatory effect of the two enzymes to increase the efficiency of the base-capture step of the other enzyme by influencing the melting of the junction base-pair of the DNA fork. The authors also use 2-aminopurine studies to precisely position the two enzymes at the DNA fork and probe the number of base-pairs each enzyme is able to influence to melt or “breath” at the fork junction. In line with the track record from this group the experiments are of high quality, yet some points should be addressed. Most comments require only clarification of the text, but some additional experiments, which should not take much time, are needed.

Issues that require additional experimentation:

1) In Figures 1 and 2, a single length of dsDNA is used to estimate unwinding parameters. In the past, this group has made use of different lengths of dsDNA to determine the kinetic parameters. The subsequent data are used to extract the effect of different GC composition and dNTP pools. This becomes important especially with the DNAP data collected in the presence of SSB where, in addition to a lag, the traces show a clear biphasic behavior. The authors chose to analyze only the first phase. How is this signal amplitude correlated to the fraction of dsDNA unwound? This parameter appears to be in Equations 1-3. The authors offer no proof or explanation that the latter can simply be ignored as they have done.

Data at different lengths of the dsDNA should be collected to show that the estimates from a single length are robust. This becomes important when using SSB which shows multi-phasic behavior and for which the authors do not describe how the fluorescence data have been calibrated to yield the fraction of unwound product.

2) SSB is used as a tool to amplify the poor strand displacement of DNAP. Given that the authors' aim is to determine the functional coupling of the T7 system, it is not clear (aside from the fact that higher concentrations would be needed) why gp2.5 was not used rather than SSB. Is the GC sensitivity of DNAP altered with gp2.5? Does gp2.5 show the same biphasic behavior in signal as SSB? A couple of selected experiments with gp2.5 would be appropriate.

3) The kinetic unwinding data and the 2-AP date are disjoined and the manuscript reads almost as two independent works. Both sets of data are absolutely critical to the overall model and the authors need to do a better job in both sections to tie the data to each other and the model. Figure 5 and Figure 6 are difficult to follow and they seem to repeat some of the same data. These figures should be re-organized to improve clarity. For example, based on the cartoons in Figure 5 and Figure 6 this appears to be the same DNA with the data comparing the DNAP only, helicase only and DNAP+Helicase cases. These data would be clearer and better summarized with a single cartoon and histogram with DNA, DNA+DNAP, DNA+helicase, DNA+DNAP+Helicase.

Also, the authors do not provide data for DNAP+SSB or gp2.5. If it is true that the function of SSB is to prevent re-annealing then SSB should not have an effect on the observed bp opening of DNAP.

Issues that require only clarification of the presentation:

4) The authors should stress that the stimulatory effect of each protein on the other is mediated through the influence of melting the junction base-pairs at the DNA fork to enhance the base-capture step rather than using language that suggests a classical protein-protein interaction; for example in the Abstract: “…helicase stimulates DNAP by promoting dNTP binding, DNAP stimulates the helicase by increasing the unwinding rate constant,…”.

Discussion on this point could benefit from the work reported by Thomen, Lopez, Bockelmann, Guilerez, Dreyfus, & Heslot in (2008) Biophysical Journal, 95:2423-2433 likening, in this case, a closed junction base-pair to a competitive inhibitor of DNAP and a non-competitive inhibitor of helicase; it would make the effect on *K*_m_ and *k*_cat_ easier to understand for the reader.

5) The authors provide little or no explanation for the large synergistic increase in 2-aminopurine fluorescence when both helicase and DNAP are present on the forked-DNA substrate above and beyond the effect of either enzyme alone. Discussion of base pair melting a kin to “breathing” fluctuations at the DNA junction would be useful to explain that the two enzymes together shift the equilibrium of the junction base pairs even more to the open/melted state than either enzyme alone, thereby producing a synergistic/larger fluorescence. Might the 2-aminopurine studies suggest that DNAP likely influences the melting of three bases at the fork junction instead of only two bases?

6) The authors inaccurately discuss the conclusions and model presented in [29]. [29] presents a similar coupling/stimulation of helicase and DNAP that is mediated through the DNA and also predicts that helicase is closer to the fork junction than DNAP. Like this manuscript, [29] attributes the stimulation of the helicase to destabilization of the first few base pairs of the DNA fork by the DNAP. Unlike this manuscript, [29] views the helicase as releasing the fork regression pressure on the DNAP thus allowing the DNAP to remain in the polymerization conformation and not switch to the exonuclease conformation as the cause of the stimulation of the DNAP. The authors should make this clarification in the revised manuscript. Can the authors comment on the possible implications of T7 DNAP exonuclease activity on their model?

---

## [Author Response]

*Issues that require additional experimentation*:

*1) In*
Figures 1 and 2*, a single length of dsDNA is used to estimate unwinding parameters. In the past, this group has made use of different lengths of dsDNA to determine the kinetic parameters. The subsequent data are used to extract the effect of different GC composition and dNTP pools. This becomes important especially with the DNAP data collected in the presence of SSB where, in addition to a lag, the traces show a clear biphasic behavior*.

Data at different lengths of the dsDNA should be collected to show that the estimates from a single length are robust. This becomes important when using SSB which shows multi-phasic behavior and for which the authors do not describe how the fluorescence data have been calibrated to yield the fraction of unwound product.

The reviewers point out that the biphasic nature of the fluorescence unwinding data with SSB and DNAP necessitates experiments with different DNA lengths to show that rate estimates from one length are reliable. This would be similar to our previous analysis of length dependence experiments with the helicase (Donmez and Patel. EMBO, 2008). The revised manuscript addresses this and we now include the fluorescence trace with two lengths of dsDNA in Figure 1—figure supplement 1. The fluorescence traces with 25 bp and 40 bp duplex DNAs show increased lag time with increased dsDNA length.

However, we noted that the rise in the kinetics becomes shallower as the dsDNA length increases, which suggests broader distribution of rates on the longer dsDNA, which makes sense because the opportunity to stall increases with dsDNA length. Therefore, it is important to keep the dsDNA constant while measuring rates as a function of increasing GC% as we have in this study.

*How is this signal amplitude correlated to the fraction of dsDNA unwound? This parameter appears to be in Equations 1-3. The authors offer no proof or explanation that the latter can simply be ignored as they have done. The authors chose to analyze only the first phase*.

It is difficult to estimate the exact amount of dsDNA unwound from the fluorescence amplitude. Therefore, we do not extract this information and simply fit the lag kinetics and exponential rise to determine the average rate of unwinding. As noted, we see two phases and the rate of the fast phase matches closely with the strand displacement rate of T7 DNAP with SSB obtained from our gel based assays (36), which indicates that unwinding and synthesis indeed takes place during the fast phase. We wondered about the slow phase and carried out many control experiments. Increasing *E. coli* SSB concentration resulted in reduced slow phase amplitude but not complete elimination (Figure 1—figure supplement 2). We speculate that the slow phase represents a population that unwinds the dsDNA at an overall slower rate not assisted by SSB.

*2) SSB is used as a tool to amplify the poor strand displacement of DNAP. Given that the authors' aim is to determine the functional coupling of the T7 system, it is not clear (aside from the fact that higher concentrations would be needed) why gp2.5 was not used rather than SSB*.

There are several reasons we use *E. coli* SSB instead of T7 gp2.5. First, SSB has higher affinity for ssDNA as compared to gp2.5 (SSB *K*_d_ of 0.1-10 nM vs low μM *K*_d_ of T7 gp2.5 complex), which means we need lower concentrations of SSB in our stopped-flow experiments and lower fluorescence background. Second, we wanted to use a trap that would not interact with T7 DNAP and T7 gp2.5 binds specifically to T7 DNAP. Also note that *E. coli* SSB is not a physiological irrelevant protein, as it is present in the cell during T7 phage infection of *E. coli*, hence it is a host factor that is present in abundant quantities and may play a role in T7 replication, but this has not been explored.

*Is the GC sensitivity of DNAP altered with gp2.5? Does gp2.5 show the same biphasic behavior in signal as SSB? A couple of selected experiments with gp2.5 would be appropriate*.

We measured the unwinding rates of T7 DNAP with 50% GC fork at different dNTP concentrations with gp2.5 and *E. coli* SSB (see Figure 7) and obtain similar dNTP *K*_m_ and unwinding *k*_cat_. We also observed biphasic kinetics of unwinding with gp2.5.

Author response image 1.**DOI:**
http://dx.doi.org/10.7554/eLife.06562.023

*3) The kinetic unwinding data and the 2-AP date are disjoined and the manuscript reads almost as two independent works. Both sets of data are absolutely critical to the overall model and the authors need to do a better job in both sections to tie the data to each other and the model*.

We agree the kinetics provides functional and 2-AP experiments structural information and both are necessary to derive the final model shown in Figure 6. We use the Discussion section to connect the functional and structural information rather than the Results section. We have now edited both sections to make the Results and Discussion sections more precise.

Figure 5
*and*
Figure 6
*are difficult to follow and they seem to repeat some of the same data. These figures should be re-organized to improve clarity. For example, based on the cartoons in*
Figure 5
*and*
Figure 6
*this appears to be the same DNA with the data comparing the DNAP only, helicase only and DNAP+Helicase cases. These data would be clearer and better summarized with a single cartoon and histogram with DNA, DNA+DNAP, DNA+helicase, DNA+DNAP+Helicase*.

We thank the reviewers for their suggestion. We have combined the two figures (new Figure 5) and show control experiments as supplementary figures.

Also, the authors do not provide data for DNAP+SSB or gp2.5. If it is true that the function of SSB is to prevent re-annealing then SSB should not have an effect on the observed bp opening of DNAP.

We agree with the reviewers that 2-AP experiments with SSB will add to the manuscript and we performed these experiments with both helicase and DNAP. SSB by itself does not significantly increase the 2-AP fluorescence of junction base pair, consistent with the idea that SSB does not melt the DNA. Interestingly, the presence of SSB enhances the 2-AP fluorescence increase by DNAP but not by helicase.

We explain the results as follows: The fluorescence of 2-AP at steady state is dictated by the equilibrium distribution of base-pair in melted and annealed states, as pointed out by the reviewers (see below). The smaller change in 2-AP intensity with DNAP alone indicates that the equilibrium is more toward the base pair annealed state. The higher 2-AP intensity with SSB indicates that equilibrium is now shifted to the base-pair separated state. Because SSB itself does not unwind DNA, this means that SSB aids the DNAP by preventing base pair re-annealing. Interestingly, SSB does not increase the 2-AP intensity when added to T7 helicase, which is consistent with our results that SSB does not aid the helicase and does not affect the unwinding rates of the helicase.

These data are now included in Figure 5—figure supplement 3 and Figure 5—figure supplement 5, and we have added the following lines to the Results section:

“When 2-AP experiments with T7 DNAP were carried out with *E. coli* SSB, the fluorescence intensity changes were much larger, although SSB by itself did not increase the 2-AP fluorescence significantly (Figure 5—figure supplement 3). These results indicate that T7 DNAP on its own only partially melts the junction base pairs. Furthermore, these base pairs appear to be in dynamic equilibrium (closed <=> open), because SSB can shift the equilibrium towards the open state by simply binding to ssDNA. This provides direct proof that T7 DNAP can melt the junction base pair, but DNAP is not efficient at preventing base pair reannealing; thus, SSB stimulates the activity of DNAP by trapping the unwound bases.”

And

“Interestingly, SSB has no effect on the helicase catalyzed melting of the fork junction (Figure 5—figure supplement 5). This is consistent with the observation that SSB does not stimulate the unwinding rates of the helicase (8).”

*Issues that require only clarification of the presentation*:

*4) The authors should stress that the stimulatory effect of each protein on the other is mediated through the influence of melting the junction base-pairs at the DNA fork to enhance the base-capture step rather than using language that suggests a classical protein-protein interaction; for example in the Abstract: “…helicase stimulates DNAP by promoting dNTP binding, DNAP stimulates the helicase by increasing the unwinding rate constant,…”*.

*Discussion on this point could benefit from the work reported by Thomen, Lopez, Bockelmann, Guilerez, Dreyfus, & Heslot in (2008) Biophysical Journal, 95:2423-2433 likening, in this case, a closed junction base-pair to a competitive inhibitor of DNAP and a non-competitive inhibitor of helicase; it would make the effect on* K_*m*_
*and* k_*cat*_
*easier to understand for the reader.*

We thank the reviewers for the suggestion. We have changed the Abstract to read:

“The synergistic shift in equilibrium of junction base-pair melting by combined enzymes explains the cooperativity, wherein helicase stimulates the polymerase by promoting dNTP binding (decreasing dNTP *K*_m_), polymerase stimulates the helicase by increasing the unwinding rate-constant (*k*_cat_), consequently the combined enzymes unwind DNA with kinetic parameters resembling enzymes translocating on single-stranded DNA.”

And have included the following sentence in the Results section:

“T7 DNAP follows an ordered mechanism, wherein dNTP binding follows the base-capture step. Therefore, the kinetic outcome of increasing GC content is analogous to a pure competitive mechanism where inhibitor (GC content) increases the dNTPs *K*_m_ without affecting the unwinding *k*_cat_. T7 helicase does not follow an ordered mechanism. Consequently, the kinetic outcome of increasing GC content is analogous to a mixed inhibition mechanism where inhibitor (GC content) decreases the unwinding *k*_cat_ and mildly affects the dTTP *K*_m_.”

*5) The authors provide little or no explanation for the large synergistic increase in 2-aminopurine fluorescence when both helicase and DNAP are present on the forked-DNA substrate above and beyond the effect of either enzyme alone. Discussion of base pair melting a kin to “breathing” fluctuations at the DNA junction would be useful to explain that the two enzymes together shift the equilibrium of the junction base pairs even more to the open/melted state than either enzyme alone, thereby producing a synergistic/larger fluorescence*.

We thank the reviewers for pointing this out. We have included the reviewers’ suggestion in the manuscript. In particular, in the subsection “T7 DNAP unwinds two base pairs and interacts with three nucleotides on the template”, we now say:

“Furthermore, these base pairs appear to be in dynamic equilibrium (closed <=> open), because SSB can shift the equilibrium towards the open state by simply binding to ssDNA. This provides direct proof that T7 DNAP can melt the junction base pair, but DNAP is not efficient at preventing base pair reannealing; thus, SSB stimulates the activity of DNAP by trapping the unwound bases.”

And in the subsection headed “Synergistic melting of junction base pairs by T7 helicase and T7 DNAP and their precise positions at the fork junction” we say:

“The 2-AP fluorescence intensity at steady state measures the equilibrium distribution of melted and annealed states of the junction base-pair. The small increase in fluorescence intensity with helicase and DNAP suggests that each enzyme shifts the equilibrium only moderately to the base-pair melted state. On the other hand, the striking increase in fluorescence intensity with the combined enzymes indicates that together the two enzymes shift the equilibrium strongly toward the base-pair melted state.”

Might the 2-aminopurine studies suggest that DNAP likely influences the melting of three bases at the fork junction instead of only two bases?

We assume the reviewers’ suggestion comes from the observation that there is a synergistic increase in fluorescence when the probe is at the N+3 position (Figure 5). Although this may be indicative of melting of the N+3 base, we think the increase observed here is due to loss of stacking interactions from melting of the N+2 base pair. We make this conclusion because synergistic melting by the combined enzymes is not observed when 2-AP is part of the junction base pair in the N+3 position (Figure 5—figure supplement 2). We now clarify this in the manuscript and say:

“Similarly, when 2-AP is at N+2 and is part of the fork junction, addition of T7 DNAP increases the fluorescence (Figure 5—figure supplement 2). However, when the 2-AP is at N+3 and is part of the fork junction, no increase in fluorescence is observed (Figure 5—figure supplement 2), which indicates that T7 DNAP does not melt the junction base pair three nucleotides downstream from the primer-end. The increase in 2-AP intensity at N+3 in the internal position (Figure 5) is due to N+2 base unstacking and not from unstacking of the N+3 base. Taken together these results indicate that T7 DNAP binds to three template-bases downstream of primer-end and melts two base pairs.”

*6) The authors inaccurately discuss the conclusions and model presented in*
[29]*.*
[29]
*presents a similar coupling/stimulation of helicase and DNAP that is mediated through the DNA and also predicts that helicase is closer to the fork junction than DNAP. Like this manuscript,*
[29]
*attributes the stimulation of the helicase to destabilization of the first few base pairs of the DNA fork by the DNAP. Unlike this manuscript,*
[29]
*views the helicase as releasing the fork regression pressure on the DNAP thus allowing the DNAP to remain in the polymerization conformation and not switch to the exonuclease conformation as the cause of the stimulation of the DNAP. The authors should make this clarification in the revised manuscript.*

We thank the reviewers for the clarification. We have made the appropriate changes in the Introduction.

Can the authors comment on the possible implications of T7 DNAP exonuclease activity on their model?

The same GC dependencies were observed with the exonuclease proficient T7 DNAP (data not included in the manuscript). As the GC content was increased, the *K*_m_ increased but the *k*_cat_ was moderately affected. Thus, the model holds true for both exo+ and exo- DNAPs.